

# GeoChronR – an R package to model, analyze and visualize age-uncertain paleoscientific data

Nicholas McKay[1], Julien Emile-Geay[2], and Deborah Khider[3]

[1]School of Earth and Sustainability, Northern Arizona University, Flagstaff, AZ 86011
[2]Department of Earth Sciences, University of Southern California, Los Angeles, CA, 90089
[3]Information Sciences Institute, University of Southern California, Marina del Rey, CA

**Correspondence:** Nicholas McKay (Nicholas.McKay@nau.edu)

**Abstract.** Chronological uncertainty is a hallmark of the paleosciences. While many tools have been made available to researchers to quantify age uncertainties suitable for various settings and assumptions, disparate tools and output formats often discourage integrative approaches. In addition, associated tasks like propagating age model uncertainties to subsequent analyses, and visualizing the results, have received comparatively little attention in the literature and available software. Here we

describe GeoChronR, an open-source R package to facilitate these tasks. GeoChronR is built around emerging data standards for the paleosciences (Linked PaleoData, or LiPD), and offers access to four popular age modeling techniques (Bacon, BChron, Oxcal, BAM). The output of these models is used to support ensemble-aware analyses, quantifying the impact of chronological uncertainties on common analyses like age-uncertain correlation, regression, principal component, and spectral analyses. We present five real-world use cases to illustrate how GeoChronR may be used to facilitate these tasks, to visualize the results in

intuitive ways, and to store the results for further analysis, promoting transparency and reusability.

## 1 Introduction

### 1.1 Background

Quantifying chronological uncertainties, and how they influence the understanding of past changes in Earth systems, is a unique and fundamental challenge of the paleosciences. Without robust error determination, it is impossible to properly assess

the extent to which past changes occurred simultaneously across regions, accurately estimate rates of change or the duration of abrupt events, or attribute causality – all of which limit our capacity to apply paleoscientific understanding to modern and future processes. The need for better solutions to both characterize uncertainty, and to explicitly evaluate how age uncertainty impacts the interpretation of records of past climate, ecology or landscapes, has been long recognized (e.g., Noren et al., 2013; National Academies of Sciences, Engineering, and Medicine, 2020). In response to this need, the paleoscience community has

made substantial advances toward improving geochronological accuracy by:

1. Improving analytical techniques that allow for more precise age determination on smaller and context-specific samples (e.g., Eggins et al., 2005; Santos et al., 2010; Zander et al., 2020)



2. Refining our understanding of how past changes in the Earth system impact chronostratigraphy, for example: improvements to the radiocarbon calibration curve (Reimer et al., 2011, 2013, 2020) and advances in our understanding of spatial variability in cosmogenic production rates used in exposure dating (Balco et al., 2009; Masarik and Beer, 2009; Charreau et al., 2019).

3. Dramatic improvement in the level of sophistication and realism in age-depth models used to estimate the ages of sequences between dated samples (e.g. Parnell et al., 2008; Bronk Ramsey, 2009; Blaauw, 2010; Blaauw and Christen, 2011).

Over the past 20 years, these advances have been widely adopted in the paleosciences, albeit partially so. Indeed, despite the progress made in quantifying uncertainty in both ages determinations and age models, few studies have formally evaluated how

chronological uncertainty may have affected the inferences made from them. For instance, whereas the algorithms mentioned above have been broadly used, studies typically calculate a single "best" estimate (often the posterior median or mean), use this model to place measured paleoclimatic or paleoenvironmental data on a timescale, and then proceed to analyze the record with little to no reference to the uncertainties generated as part of the age modeling exercise, however rigorous in its own right. In addition, few studies have evaluated sensitivity to the choice of age modeling technique or choice of parameters, so that the

typical discussion of chronological uncertainties remains partial and qualitative.

This paradigm is beginning to change. In recent years, some studies have taken advantage of approaches that generate ensembles of age models to evaluate how the results of their analyses and conclusions vary given differences between ensemble members (e.g., Tierney et al., 2013; Khider et al., 2014; Deininger et al., 2017; Khider et al., 2017; McKay et al., 2018; Bhattacharya and Coats, 2020). By thus applying an analysis to all members of an age ensemble, the precise impact of age

uncertainty may be formally evaluated.

Despite its potential to substantially improve uncertainty quantification for the paleosciences, this framework is not widely utilized. The majority of studies utilizing this approach have been regional (e.g., Tierney et al., 2013; Khider et al., 2017; Deininger et al., 2017; McKay et al., 2018; Bhattacharya and Coats, 2020) or global-scale (e.g., Shakun et al., 2012; Marcott et al., 2013; Kaufman et al., 2020a) syntheses. Some primary publications of new records incorporate time-uncertain analysis

into their studies (e.g., Khider et al., 2014; Boldt et al., 2015; Falster et al., 2018), but this remains rare. We suggest that there are several reasons for the lack of adoption of these techniques:

1. For synthesis studies, the necessary geochronological data are not publicly available for the vast majority of records. Even when they are available, the data are archived in diverse and unstructured formats. Together, this makes what should be a simple process of aggregating and preparing data for analysis prohibitively time-consuming;

2. For studies of new and individual records, few tools for ensemble analysis are available, and those that are require a degree of comfort with coding languages and scientific programming that is rare among paleoscientists;





3. There is a disconnect between age-model development and time-uncertain analysis. Published approaches have utilized either simplified age-modeling approaches (e.g, Haam and Huybers, 2010; Routson et al., 2019), or specialized approaches not used elsewhere in the community (e.g., Marcott et al., 2013; Tierney et al., 2013).

Extracting the relevant data from commonly-used age-modelling algorithms, creating time-uncertain ensembles, then reformatting those data for analysis in available tools typically requires the development of extensive custom codes. GeoChronR presents an integrative approach to facilitate this work.

## 1.2 Design principles

GeoChronR was built to lower the barriers to broader adoption of these emerging methods. Thus far, GeoChronR has been primarily designed with Quaternary datasets in mind, for which a variety of chronostratigraphic methods are available: radiometric dating ($^{14}$C, $^{210}$Pb, U/Th), exposure dating, layer-counting, flow models (for ice cores), orbital alignment, and more. Nevertheless, the primary uncertainty quantification device is age ensembles, irregardless of how they were produced. As such, GeoChronR's philosophy and methods can be broadly applicable to any dataset for which age ensembles can be generated.

GeoChronR provides an easily-accessible, open-source, and extensible software package of industry-standard and cutting-edge tools that provides users with a single environment to create, analyze, and visualize time-uncertain data. GeoChronR is designed around emerging standards in the paleosciences that connects users to growing libraries of standardized datasets formatted in the Linked PaleoData format (McKay and Emile-Geay, 2016), including thousands of datasets archived at the World Data Service for Paleoclimatology (WDS-Paleo) and lipdverse.org, those at the LinkedEarth wiki (http://wiki.linked.earth), and Neotoma (Williams et al., 2018) via the neotoma2lipd package (McKay, 2020). GeoChronR reuses existing community packages, for which it provides a standardized interface, with LiPD as input/output format. Central to the development of the code and documentation were two workshops carried out in 2016 and 2017 at Northern Arizona University, gathering a total of 33 participants. The workshop participants were predominantly early career researchers with $> 50\%$ participation of women, who are underrepresented in the geosciences. Exit surveys were conducted to gather feedback, and to suggest improvements and extensions, which were integrated into subsequent versions of the software.

## 1.3 Outline of manuscript

This manuscript describes the design, analytical underpinnings, and most common use cases of GeoChronR. Section 2 describes the integration of age modelling algorithms with GeoChronR. Section 3 details the methods implemented for age uncertain analysis. Section 4 goes through the principles and implementation of age-uncertain data visualization in GeoChronR, and section 5 provides five real-world examples of how GeoChronR can be used for paleoscientific workflows.

## 2 Age uncertainty quantification in GeoChronR

GeoChronR does not introduce any new approaches to age uncertainty quantification; rather, it integrates existing, widely-used packages while streamlining the acquisition of age ensemble members. Fundamentally, there are two types of age models used





in the paleosciences: tie-point and layer-counted. Most of the effort in age uncertainty quantification in the community has been focused on tie-point modelling, where the goal is to estimate ages (and their uncertainties) along a depth profile given chronological estimates (and their uncertainties) at multiple depths downcore. Over the past 20 years, these algorithms have progressed from linear or polynomial regressions with simple characterizations of uncertainty (Heegaard et al., 2005; Blaauw,

2010) to more rigorous techniques, particularly Bayesian approaches: as of writing, the three most widely used algorithms are Bacon (Blaauw and Christen, 2011), BChron (Parnell et al., 2008), and OxCal (Bronk Ramsey, 2008), which are all Bayesian age-deposition models that estimate posterior distributions on age-depth relationships using different assumptions and methodologies. Trachsel and Telford (2017) reviewed the performance of these three algorithms, as well as a non-Bayesian approach (Blaauw, 2010), and found that the three Bayesian approaches generally outperform previous algorithms, especially

when appropriate parameters are chosen (although choosing appropriate parameters can be challenging). Bacon, BChron and Oxcal all leverage Monte Carlo Markov Chain (MCMC) techniques to sample the posterior distributions, thereby quantifying age uncertainties as a function of depth in the section. GeoChronR interfaces with each of these algorithms through their R packages (Parnell et al., 2008; Blaauw et al., 2020; Martin et al., 2018), standardizing and streamlining the input and the extraction of the age ensembles from the MCMC results for further analysis.

In addition to working with ensembles from tie-point age models, GeoChronR connects users to probabilistic models of layer-counted chronologies. BAM (Comboul et al., 2014) was designed to probabilistically simulate counting uncertainty in banded archives, such as corals, ice cores, or varved sediments, but can be used to crudely simulate age uncertainty for any record, and is useful when the data or metadata required to calculate an age-depth model are unavailable (e.g. Kaufman et al., 2020a). Here we briefly describe the theoretical basis and applications of each of the four approaches integrated in GeoChronR.

## 2.1  Bacon

The Bayesian ACcumulatiON (Bacon) algorithm (Blaauw and Christen, 2011) is one of the most broadly used age-modelling techniques, and was designed to take advantage of prior knowledge about the distribution and autocorrelation structure of sedimentation rates in a sequence to better quantify uncertainty between dated levels. Bacon divides a sediment sequence into a parameterized number of equally-thick segments; most models use dozens to hundreds of these segments. Bacon then

models sediment deposition, with uniform accumulation within each segment, as an autoregressive gamma process, where both the amount of autocorrelation and the shape of the gamma distribution are given prior estimates. The algorithm employs an adaptive Markov Chain Monte Carlo algorithm that allows for Bayesian learning to update these variables given the age-depth constraints, and converge on a distribution of age estimates for each segment in the model. Bacon has two key parameters: the shape of the accumulation prior, and the segment length, which can interact in complicated ways (Trachsel and Telford,

2017). In our experience, the segment length parameter has the greatest impact on the ultimate shape and amount of uncertainty simulated by Bacon, as larger segments result in increased flexibility of the age-depth curve, and increased uncertainty between dated levels. Bacon is written in C++ and R, with an R interface. More recently, the authors released an R package "rbacon" (Blaauw et al., 2020), which GeoChronR leverages to provide access to the algorithm. Bacon will optionally return a subset





of the MCMC accumulation rate ensemble members with high *a posteriori* probabilities, which GeoChronR uses to form age
ensemble members for subsequent analysis.

## 2.2 BChron

BChron (Haslett and Parnell, 2008; Parnell et al., 2008) uses a similar approach, using a continuous Markov monotone stochas-
tic process coupled to a piecewise linear deposition model. This simplicity allows semi-analytical solutions that make BChron
computationally efficient. BChron was originally intended to model radiocarbon-based age-depth models in lake sedimentary
cores of primarily Holocene age, but its design allows broader applications. In particular, modeling accumulation as additive
independent gamma increments is appealing for the representation of hiatuses, particularly for speleothem records, where ac-
cumulation rate can vary quite abruptly between quiescent intervals of near-constant accumulation (Parnell et al., 2011; Dee
et al., 2015; Hu et al., 2017). The downside of this assumption is that BChron is known to exaggerate age uncertainties in cases
where sedimentation varies smoothly (Trachsel and Telford, 2017).

Bchron has several key parameters, which allow a user to encode their specific knowledge about their data. In particular,
the `outlierProbs` parameter is useful in giving less weight to chronological tie points that may be considered outliers
either because they create a reversal in the stratigraphic sequence or they were flagged during analysis (e.g. contamination).
This is extremely useful for radiocarbon-based chronologies where the top age may not be accurately measured for modern
samples. The `thetaMhSd`, `psiMhSd`, and `muMhSd` parameters control the Metropolis-Hastings standard deviation for the age
parameters and Compound Poisson-Gamma scale and mean respectively, which influence the width of the ensemble between
age control tie points. These parameters use the same default values as the official Bchron package, and we recommend that
users only change them if they have good prior reason to do so.

## 2.3 Oxcal

The OxCal software package has a long history and extensive tools for the statistical treatment of radiocarbon and other
geochronological data (Bronk Ramsey, 1995). In Bronk Ramsey (2008), age-depth modelling was introduced with three op-
tions for modelling depositional processes that are typically useful for sedimentary sequences: uniform, varve, and Poisson de-
position models, labeled U-sequence, V-sequence and P-sequence, respectively. The Poisson-based model is the most broadly
applicable for sedimentary, or other accumulation-based archives (e.g. speleothems), and although any sequence type can be
used in GeoChronR, most users should use a P-sequence, which is the default. Analogously to segment length parameter in
Bacon, the *k* parameter (called `eventsPerUnitLength` in GeoChronR), controls how many events are simulated per unit
of depth, and has a strong impact on the flexibility of the model, as well as the amplitude of the resulting uncertainty. As the
number of events increases, the flexibility of the model, and the uncertainties, decrease. Trachsel and Telford (2017) found
that this parameter has a large impact on the accuracy of the model, more so than the choices made in Bacon or Bchron.
Fortunately, Bronk Ramsey et al. (2010) made it possible for *k* to be treated as a variable, and the model will estimate the
most likely values of *k* given a prior estimate and the data. The downside of this flexibility is that this calculation can greatly
increase the convergence time of the model. Oxcal is written in C++, with an interface in R (Martin et al., 2018). Oxcal does





not typically calculate posterior ensembles for a depth sequence, but can optionally output MCMC posteriors at specified levels in the sequence. GeoChronR uses this feature to extract ensemble members for subsequent analysis.

## 2.4 Banded Age Model (BAM)

Comboul et al. (2014) is a probabilistic model of age errors in layer-counted chronologies. The model allows a flexible para-
metric representation of such errors (either as Poisson or Bernoulli processes), and separately considers the possibility of double-counting or missing a band. The model is parameterized in terms of the error rates associated with each event, which are intuitive parameters to paleoscientists, and may be estimated via replication (DeLong et al., 2013). In cases where such rates can be estimated from the data alone, an optimization principle may be used to identify a more likely age model when a high-frequency common signal can be used as a clock (Comboul et al., 2014). As of now, BAM does not consider uncer-
tainties about such parameters, representing a weakness of the method. Bayesian generalizations have been proposed (Boers et al., 2017), which could one day be incorporated into GeoChronR if the code is made public. BAM was coded in MATLAB, Python and R, and it is this latter version that GeoChronR uses.

## 2.5 Data storage

GeoChronR archives the outcome of all of these models using in the LiPD format (McKay and Emile-Geay, 2016). One of the
primary motivations for LiPD was to facilitate age-uncertain analysis, and GeoChronR is designed to leverage these capabilities. LiPD can store multiple chronologies (called "chronData" in LiPD), each of which can contain multiple measurement tables (which house the measured chronological constraints) and any number of chronological models (which comprise both the results produced of the analysis, as well as metadata about the method used to produce those results) (figure 1). In LiPD, chronological models include up to three types of tables:

1. Ensemble tables, which store the output of an algorithm that produces age model ensembles, and a reference column (typically depth),

   2. Summary tables, which describe summary statistics produced by the algorithm (e.g., median and $2\sigma$ uncertainty ranges), and

   3. Distribution tables, which store age-probability distributions for calibrated ages, typically only used for calibrated radio-
carbon ages.

LiPD can also store relevant metadata about the modeling exercise, including the values of the parameters used to generate the data tables. This storage mechanism allows for an efficient sweep over the function parameters and comparison of the results.

The capability of GeoChronR to structure the output of the popular age model algorithms described in this section into LiPD
is a key value proposition of GeoChronR. Once structured as a LiPD object in R, these data and models can be written out to a LiPD file and readily analyzed, shared and publicly archived.





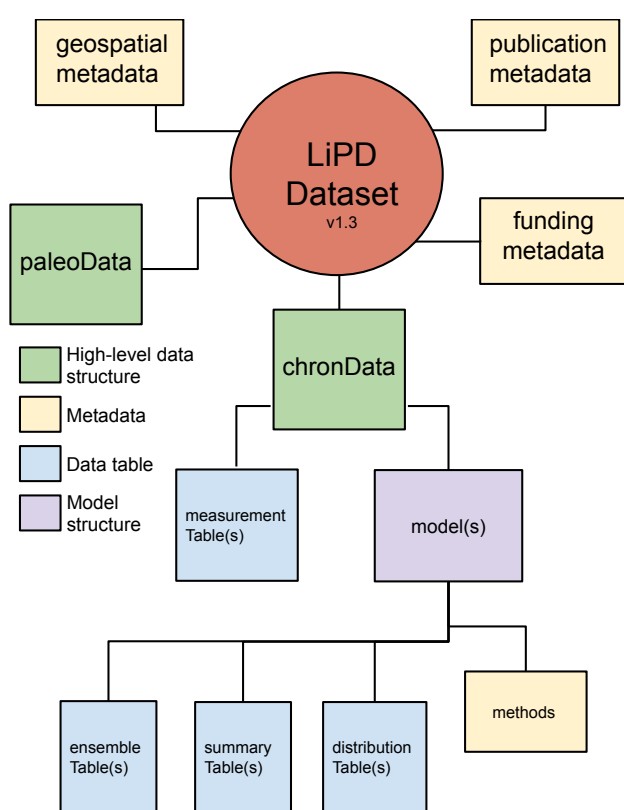

**Figure 1.** Schematic representation of a Linked PaleoData (LiPD) dataset, with a focus on chronological data. A LiPD dataset can contain one or more instances of all of the data objects and structures. The paleoData structure mirrors that of the chronData, but is not shown for clarity.

## 3 Age-uncertain data analysis in GeoChronR

Some theoretical work has attempted to quantify how chronological uncertainty may affect various paleoscientific inferences (e.g. Huybers and Wunsch, 2004); however, such efforts are hard to generalize given the variety of age-uncertainty structures in real-world paleoscientific data. Consequently, GeoChronR follows a general, pragmatic, and broadly-used approach that

5 leverages age ensembles (and optionally, ensembles of climate proxy or paleoenvironmental data), to propagate uncertainties through all steps of an analysis. Effectively, this is done by randomly sampling ensemble members and then repeating the analysis many (typically hundreds to thousands) times, each time treating a different ensemble member as "true". This builds an output ensemble that quantifies the impact of those uncertainties on a particular inference. These output ensembles rarely lend themselves to binary significance tests (e.g., a p-value below 0.05), but are readily used to estimate probability densities

10 or quantiles, and thus provide quantitative evidence for which results are robust to age and proxy uncertainty (and which are





not). Version 1.0.0 of GeoChronR has implemented ensemble analytical techniques for four of the most common analyses in the paleosciences: correlation, regression, spectral, and principal component analyses.

## 3.1 Correlation

Pearson's product-moment correlation is the most common measure of a relationship between two variables $X$ and $Y$.

Its computation is fast, lending itself to ensemble analysis, with a handful of pretreatment and significance considerations that are relevant for ensembles of paleoscientific data.

First, correlation analysis for timeseries is built on the assumption the datasets can be aligned on a common timeline. Age-uncertain data violate this assumption. We overcome this by treating each ensemble member from one or more age uncertain timeseries as valid for that iteration, then "bin" each of the timeseries into coeval intervals. The "binning" procedure

in GeoChronR sets up an interval, which is typical evenly spaced, over which the data are averaged. Generally, this intentionally degrades the median resolution of the timeseries, for example, a timeseries with 37-year median spacing could be reasonably "binned" into 100- or 200-year bins. The binning procedure is repeated for each ensemble member, meaning that between different ensembles, different observations will be placed in different bins.

Following binning, Pearson correlation is calculated and recorded for each ensemble member.

The standard way to assess correlation significance is using a Student's T-test, which assumes normality and independence. By default, GeoChronR maps both datasets to a standard normal distribution (van Albada and Robinson, 2007, Emile-Geay and Tingley (2016)), thus ensuring normality. An additional difficulty is that paleoscientific timeseries are often highly autocorrelated, leading to spurious assessments of significance (Hu et al., 2017).

GeoChronR addresses this point using three approaches:

1. The simplest approach is to adjust the test's sample size to reflect the reduction in degrees of freedom due to autocorrelation. Following Dawdy and Matalas (1964), the effective number of degrees of freedom is $\nu = n\frac{1-\phi_{1,X}\phi_{1,X}}{1+\phi_{1,X}\phi_{1,X}}$, where $n$ is the sample size (here, the number of bins) and where $\phi_{1,X}, \phi_{1,X}$ are the lag-1 autocorrelation of two time series $X, Y$, respectively. This approach is called "effective-n" in GeoChronR. It is an extremely simple approach, with no added computations by virtue of being a parametric test using a known distribution ($t$ distribution). A downside is that the correction is approximate, and can substantially reduce the degrees of freedom (Hu et al., 2017), to less than 1 in cases of high autocorrelation, which are common in paleoscientific timeseries. This may result in overly conservative assessment of significance, so this option is therefore not recommended.

    2. A parametric alternative is to generate surrogates, or random synthetic timeseries, that emulate the persistence characteristics of the series. This "isopersistent" test generates $M$ (say, 500) simulations from an autoregressive process of order 1 (AR(1)), which has been fitted to the data. These random timeseries are then used to obtain the null distribution, and compute p-values, which therefore measure the probability that a correlation as high as the one observed ($r_o$) could have arisen from correlating $X$ or $Y$ with AR(1) series with identical persistence characteristics as the observations. This





approach is particularly suited if an AR model is a sensible approximation to the data, as is often the case (Ghil et al., 2002). However, it may be overly permissive or overly conservative in some situations.

3. A non-parametric alternative is the approach of Ebisuzaki (1997), which generates surrogates by scrambling the phases of $X$ and $Y$, thus preserving their power spectrum. To generate these "isospectral" surrogates, GeoChronR uses the
`make_surrogate_data` function from the rEDM package (Park et al., 2020). This method makes the fewest assumptions as to the structure of the series, and its computational cost is moderate, making it the default in GeoChronR.

In addition to the impact of autocorrelation on this analysis, repeating the test over multiple ensemble members raises the issue of test multiplicity (Ventura et al., 2004), or the "look elsewhere effect". To overcome this problem, we control for this false discovery rate (FDR) using the simple approach of Benjamini and Hochberg (1995), coded in R by Ventura et al. (2004).
FDR explicitly controls for spurious discoveries arising from repeatedly carrying out the same test. At a 5% level, one would expect a 1000 member ensemble to contain 50 spurious "discoveries" – instances of the null hypothesis, here "no correlation" being rejected. FDR takes this effect into account to minimize the risk of identifying such spurious correlations merely on account of repeated testing. In effect, it filters the set of "significant" results identified by each hypothesis test (effective-N, isopersistent, or isospectral).

## 3.2  Regression

Linear regression is a commonly used tool to model the relationships between paleoscientific data and instrumental or other datasets. One application is calibration-in-time (Grosjean et al., 2009), whereby a proxy timeseries is calibrated to an instrumental series with a linear regression model over their period of overlap. This approach is particularly vulnerable to age uncertainties, as both the development of the relationship, and the reconstruction, are affected. GeoChronR propagates age (and
optionally proxy) uncertainties through both the fitting of the ordinary least squares regression model, and the reconstruction "forecast" using the ensemble model results and age uncertainty. Like the correlation algorithm, ensemble regression uses an ensemble binning procedure that's analogous to correlation. GeoChronR then exports uncertainty structure of the modeled parameters (e.g. slope and intercept), as well as the ensemble of reconstructed calibrated data through time.

## 3.3  Principal Component Analysis

GeoChronR implements the age-uncertain principal component analysis (PCA) procedure introduced by Anchukaitis and Tierney (2013), with some minor modifications and additions. Like correlation and regression, PCA (or empirical orthogonal function (EOF) analysis) requires temporally aligned observations, and GeoChronR uses a binning procedure to achieve this across multiple ensembles. This differs from the implementation of Anchukaitis and Tierney (2013), who interpolated the data to a common timestep. In addition, traditional singular value decomposition approaches to PCA require a complete set
of observations without any missing values. For paleoclimate data, especially when considering age uncertainty, this requirement is often prohibitive. To overcome this, GeoChronR implements multiple options for PCA analysis using the pcaMethods package (Stacklies et al., 2007). The default and most rigorously tested option is a probabilistic PCA (PPCA) approach that





uses expectation maximization algorithms to infill missing values (Roweis, 1998). This algorithm assumes that the data and their uncertainties are normally distributed, which is often (but not always) a reasonable assumption for paleoscientific data. As in the other analytical approaches in GeoChronR, users can optionally transform each series to normality using the inverse Rosenblatt transform (van Albada and Robinson, 2007). This is recommended, and is the default option, but does not guaran-
tee that the uncertainties will be Gaussian. As in correlation and regression, GeoChronR propagates uncertainties through the analysis by repeating the analysis across randomly sampled age and/or proxy ensemble members to build output ensembles of the loadings (eigenvectors), variance explained (eigenvalues) and principal component timeseries. Because the sign of the loadings in PCA analyses is arbitrary and vulnerable to small changes in the input data, GeoChronR reorients the sign of the loadings for all PCs so that the mean of the loadings is positive. For well defined modes this effectively orients ensemble PCs,
but loading orientation may be uncertain for lower order, or more uncertain, modes.

As in Anchukaitis and Tierney (2013), we use a modified version of Preisendorfer's "Rule N" (Preisendorfer and Mobley, 1988) to estimate which modes include more variability than those that can arise from random time series with comparable characteristics to the data. GeoChronR uses a rigorous "red" noise null hypothesis, modified from Neumaier and Schneider (2001), where following the selection of the age ensemble in each iterations, a synthetic autoregressive timeseries is simulated
based on parameters fit from each dataset. This means that the characteristics of the null timeseries, including the temporal spacing, autocorrelation and, optionally, the first order trend, match those of each dataset, and vary between locations and ensemble iterations. For each iteration, the ensemble PCA procedure is replicated with the synthetic null dataset, using the same age ensemble member randomly selected for the real data. This effectively propagates the impact of age uncertainty into null hypothesis testing. Following the analysis, the distribution of eigenvalues calculated by the ensemble PCA is typically
compared with the 95th percentile of the synthetic eigenvalue results in a scree plot. Only modes whose eigenvalues exceed this threshold should be considered robust.

### 3.4   Spectral Analysis

Many research questions in the paleosciences revolve around spectral analysis: describing phase leads and lags among different climate system components over the Pleistocene (e.g., Imbrie et al., 1984; Lisiecki and Raymo, 2005; Khider et al., 2017), the
hunt for astronomical cycles over the Holocene (Bond et al., 2001) or in deep time (Meyers and Sageman, 2007; Meyers, 2012, 2015; Lisiecki, 2010), or characterizing the continuum of climate variability (Huybers and Curry, 2006; Zhu et al., 2019). Yet, spectral analysis in the paleosciences faces unique challenges: chronological uncertainties, of course, as well as uneven sampling, which both violate the assumptions of classical spectral methods (Ghil et al., 2002). To facilitate the quantification of chronological uncertainties in such assessments, GeoChronR implements four spectral approaches:

1. the Lomb-Scargle periodogram (VanderPlas, 2018), which uses an inverse approach to harmonic analysis in unevenly-spaced timeseries.





2. REDFIT, a version of the Lomb-Scargle periodogram tailored to paleoclimatic data (Schulz and Mudelsee, 2002; Mudelsee, 2002; Mudelsee et al., 2009). The GeoChronR uses the implementation of REDFIT from the dplR package (Bunn, 2008).

3. the wavelet-based method of Mathias et al. (2004), called "nuspectral". This method is quite similar to the Weighted
Wavelet Z-transform algorithm of Foster (1996), though it is prohibitively slow in this implementation, and the fast version using a compact-support approximations of the mother wavelet did not perform well in our tests.

4. The multi-taper method (MTM) of Thomson (1982), a mainstay of spectral analysis (Ghil et al., 2002) designed for evenly spaced timeseries. GeoChronR uses the MTM implementation of Meyers (2014), which couples MTM to efficient linear interpolation, together with various utilities to define autoregressive and power-law benchmarks for spectral peaks.

As described in section 3.1, mapping to a standard normal is applied by default, to avoid strongly non-normal datasets from violating the methods' assumptions. This can be relaxed by setting `gaussianize = FALSE` in `computeSpectraEns`.

## 4 Visualization with GeoChronR

One of the challenges with age-uncertain analysis is that it adds at least one additional dimension to the results, which can be difficult to visualize. GeoChronR aims to facilitate simple creation of intuitive, publication-quality figures that provide mul-
tiple options for visualizing the impacts of age-uncertainty, while maintaining flexibility for users to customize their results as needed. To meet the multiple constraints of simplicity, quality and customization, GeoChronR relies heavily on the "ggplot2" package (Wickham, 2016). High-level plotting functions in GeoChronR (e.g., `plotTimeseriesEnsRibbons` and `plotPca`) produce complete figures as ggplot2 objects, that can be readily customized by adding or changing ggplot2 layers.

    The figures in the use cases (section 5) are all produced by GeoChronR and generally fall into a few categories: time-
series, mapping, and periodograms. The default graphical mode is used through the figures of this paper; this aesthetic is what GeoChronR produces by default, but is readily modified by the user as desired.

### 4.1 Timeseries

The most common figure that users produce with GeoChronR are ensemble timeseries. GeoChronR uses two complementary approaches to visualize these ensembles. The first is the simplest, where a large subset of the ensemble members are plotted
as semi-transparent lines. This approach, implemented in `plotTimeseriesEnsLines`, provides a faithful representation of the data, while the overlapping semi transparency provides a qualitative sense of the ensemble uncertainty structure. The second approach uses contours to more rigorously visualize the structure of the time-value uncertainty space represented by the ensembles. `plotTimeseriesEnsRibbons` shows the quantiles of the ensembles at specified levels as shaded bands. This approach provides the quantitative uncertainty structure, but tends to smooth out the apparent temporal evolution of the data.
Fortunately, the two approaches are complementary, and often the best approach is to quantify the ensemble distribution with





ribbons in the background, and then overlap them with a handful of ensemble lines to illustrate the structure in representative ensemble members.

## 4.2 Maps

GeoChronR has simple mapping capabilities built in that rely on the maps (Becker et al., 2018) and ggmap (Kahle and Wick-
ham, 2013) packages. The `mapLipd` and `mapTs` functions provide quick geospatial visualization of one or more datasets, but also serve as the basis for the visualization of ensemble spatial data produced by ensemble PCA analyses. In paleoscientific studies, the loadings (eigenvectors) of a PCA analysis are often portrayed as dots on a map, with a colorscale that highlights the sign and amplitude of the loadings. In ensemble PCA, the additional dimension of uncertainty in the loadings needs to be visualized as well. In GeoChronR, the median of the loadings is shown as a color, and the size of the symbol is inversely
proportional to the spread of uncertainty across the ensemble. Consequently, large symbols depict loadings that are robust to the uncertainties, whereas small symbols show datasets whose loadings change substantially across the analysis. An example is shown in section 5.4.

## 4.3 Spectra

It is customary to plot spectra on a log-log scale, which helps separate the low powers and low frequencies. This choice also
naturally highlights scaling laws (Lovejoy and Schertzer, 2013; Zhu et al., 2019) as linear structures in this reference frame. GeoChronR implements this convention by default, although the scales can be readily modified using ggplot2. In addition, the abscissa ($\log_{10} f$) is labeled according to the corresponding period, which are more intuitive than frequency to scientists reading the plot. To help identify significant periodicities, confidence limits can be superimposed, based on user-specified benchmarks (see 5). The `plotSpectrum` function visualizes single ensemble members (e.g. a median age model), while
`plotSpectraEns` visualizes the quantiles of a distribution of age-uncertain spectra as ribbons, using the eponymous ggplot2 function. `periodAnnotate` allows to manually highlight periods of interest, layered onto an existing plot.

## 5   Use cases

We now illustrate the use of these tools on five use cases. The first example shows how a user might create age ensembles on different archives, and how to visualize the timing of abrupt events with appropriate uncertainty quantification. The sec-
ond example walks through ensemble correlation of age-uncertain records. The third introduces the topic of age-uncertain calibration-in-time. The fourth provides an example of regional age-uncertain principal components analyses, and the fifth deals with spectral analysis. The complete details needed to reproduce these use cases are available in the R Markdown source code for this manuscript, and are elaborated upon with additional detail and customization options in the "vignettes" included within the GeoChronR package, as well at http://lipdverse.org/geochronr-examples/.





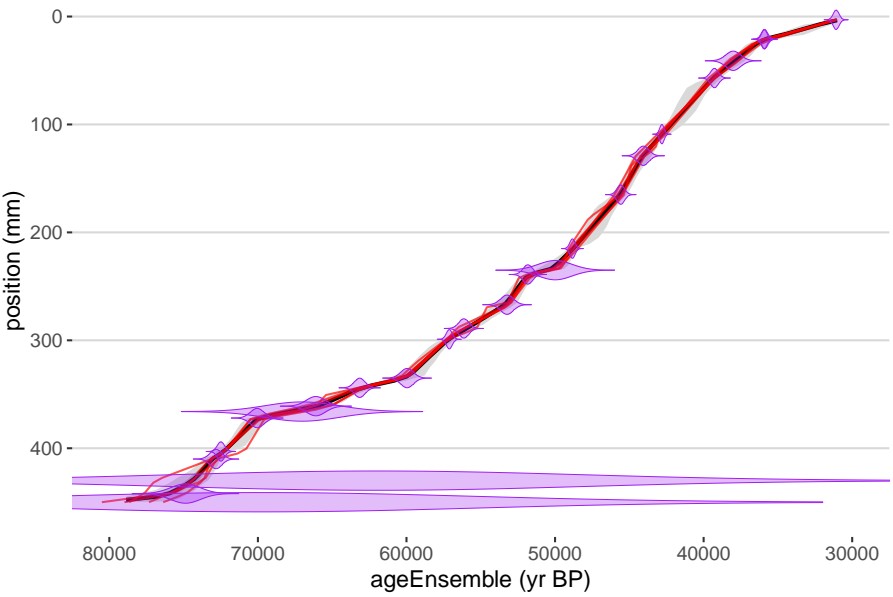

**Figure 2.** Age-depth model generated by BChron for a speleothem from Hulu Cave. Relative age probability distributions shown in purple. Median age-depth in black, with 50 and 95 percentile highest-density probability ranges shown in dark and light gray, respectively. Five random age-depth ensemble members shown in red.

## 5.1 Creating an age ensemble

A common first task when using GeoChronR is to create an age ensemble, either because the user is developing a new record, or because the age ensemble data for the record they are interested in is unavailable. As described in section 2, workflows for four published age quantification software packaged are integrated into GeoChronR. All four methods can be used simply

in GeoChronR with a LiPD file loaded into R that contains the chronological measurements, and the high-level functions `runBacon`, `runBchron`, `runOxcal` and `runBam`. These functions take LiPD objects as inputs, and return updated LiPD objects that include age-ensemble data generated by the respective software packages, with these data stored in the appropriate tables described in section 2.5. Typically, additional information (e.g., reservoir age correction) is needed to optimally run the algorithms. When these inputs are not specified, GeoChronR will run in interactive mode, asking the user which variables and

other input values they would like to use in their model. These input choices are printed to the screen while the program runs, or are available later with the function `getLastVarString`. By specifying these inputs, age model creation can be scripted and run in non-interactive mode. In this use case, we'll use GeoChronR and BChron (Parnell et al., 2008) to calculate an age ensemble for the Hulu Cave $\delta^{18}$O speleothem record (Wang et al., 2001), and BAM (Comboul et al., 2014) to simulate age uncertainties for the GISP2 ice core $\delta^{18}$O dataset (Alley, 2000). The `plotChronEns` function will plot an age-depth model

and uncertainties derived from the age ensemble (figure 2).




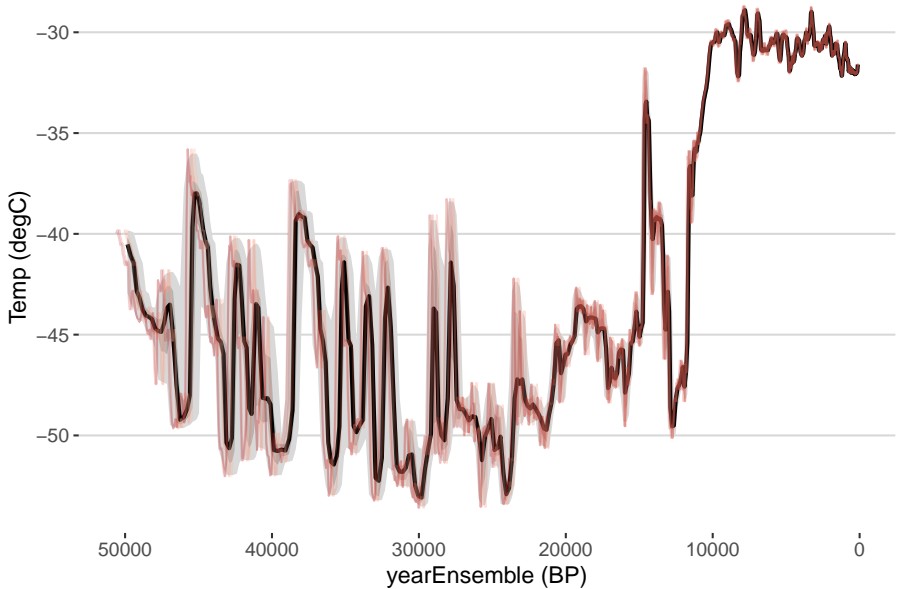

**Figure 3.** Impact of age uncertainty on reconstructed temperature at GISP2 over the past 50,000 years. The median ensemble member is shown in black, with the 50 and 95% highest-density probability ranges shown in dark and light gray, respectively. Five random age-uncertain temperature ensemble members shown in red.

After an age ensemble has been added to a LiPD object, the user can visualize the ensemble timeseries using the `plotTimeseriesEnsRibbons` and `plotTimeseriesEnsLines` functions. GISP2 $\delta^{18}$O is plotted with age uncertainty, using both functions, in figure 3.

## 5.2 Abrupt climate change in Greenland and China

Now that the user has generated age ensembles for the two datasets, they wish to see if a correlation between the two datasets is robust to the age uncertainty modeled here. On multi-millennial timescales, the two datasets display similar features, so much so that the well-dated Hulu Cave record, and other similar records from China and Greenland, have been used to argue for atmospheric teleconnections between the regions and support the independent chronology of GISP2 (Wang et al., 2001). In this use case, we revisit this relation quantitatively, and use the age models created above, as well as GeoChronR's `corEns`

function, to calculate the impact of age uncertainty on the correlation between these two iconic datasets. Note that this approach is, in many ways, simplistic. Correlating the two age-uncertain datasets will characterize the relationship, but ignores ancillary evidence that may support a mechanistic relationship between two timeseries. Still, it illustrates how age uncertainty can affect apparent alignment between two datasets, which is the purpose of this example.

  Here we calculate correlations during the period of overlap in 200 year steps, determining significance for each pair of

ensemble members while accounting for autocorrelation. GeoChronR includes four built-in approaches to estimate the sig-



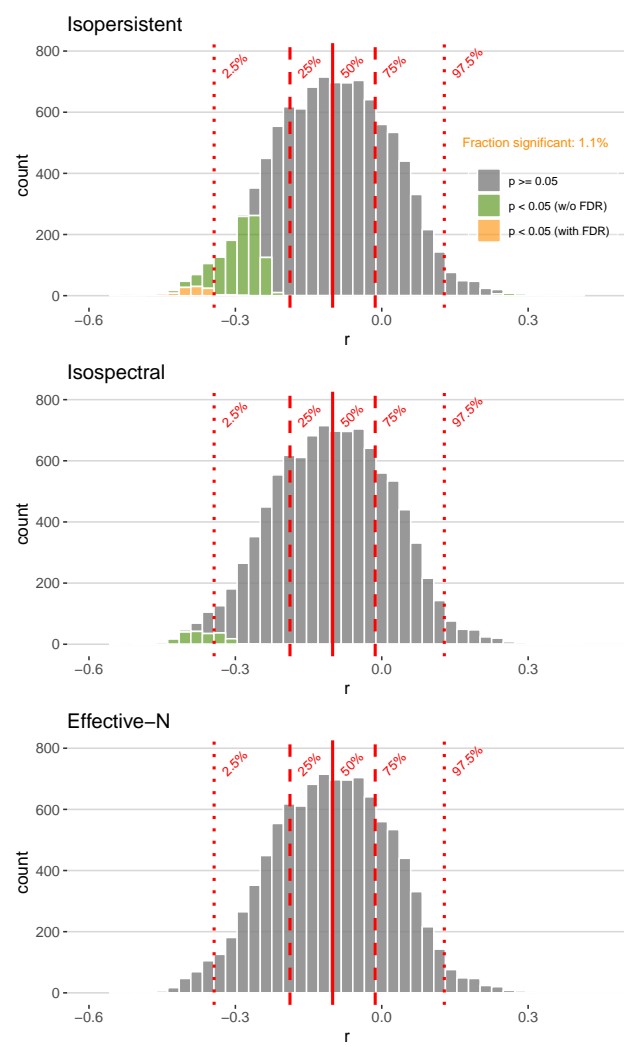

**Figure 4.** Distribution of age-uncertain correlation between Hulu Cave speleothem and GISP2 ice core $\delta^{18}$O. Significant (insignificant) correlations shown in green (gray). Correlations that remain significant after adjusting for False Discovery Rate (FDR) shown in orange. Quantiles of the distribution shown in vertical red lines.





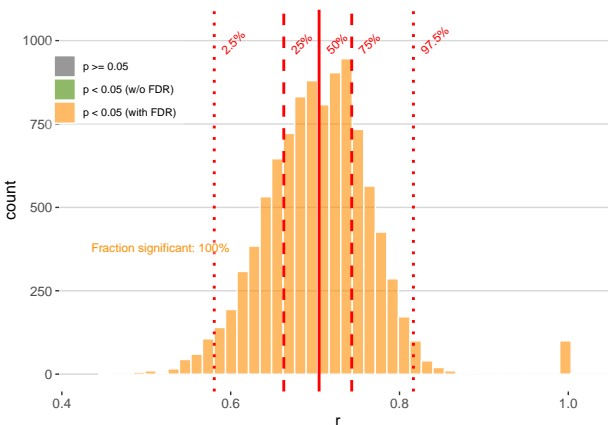

**Figure 5.** Distribution of age-uncertain correlation between Hulu Cave speleothem $\delta^{18}$O and itself, treating the age uncertainty as if independent. Conventions as in figure 4.

nificance of correlations (section 3.1). Here we examine the correlation results as histograms, with color shading to highlight significance, for each of the three methods that include a correction for autocorrelation (figure 4). The r-values are the same for all the results, only the assessment of significance changes. The two timeseries exhibit consistently negative correlations, although 21.8% of the ensemble members are positive. In this example, the isopersistent approach finds the most significant

correlations, with 12.4% signficant ensemble members, and 1.1% remain significant after adjusting for false discovery rate. In this instance, the isospectral approach is more conservative, with only 2.0% significant members, none of which remain signficant with FDR. Lastly, the effective sample size approach of Dawdy and Matalas (1964) is most conservative, finding no significant correlations.

    As mentioned above, there are many reasons to believe that there are teleconnections that link Greenland temperatures to the

dynamic monsoon circulation in Asia, especially during abrupt climate changes during glacial periods (e.g. Liu et al., 2013; Duan et al., 2016; Zhang et al., 2019). This simple correlation exercise does not affirm such a link, and we offer several reason why this may be the case.

    We first note that evaluating the significance of age uncertain correlation remains somewhat subjective, as there is no theoretical justification for what fraction of ensemble correlation results should be expected to pass such a significance test. Indeed,

two correlated timeseries, when afflicted with age uncertainty, will commonly return some fraction of insignificant results when random ensemble members are correlated against each other. The frequency of these "false negatives" depends on the structure of the age uncertainties and the timeseries, and will vary to some extent by random chance. One way to get a sense of the vulnerability of a timeseries to false negatives is to perform an age-uncertain correlation of a dataset with itself. It is appropriate to consider the results of this analysis as a best-case scenario, and to consider the correlation results in this light.

For illustration, we perform this calculation with the Hulu Cave $\delta^{18}$O record (figure 5).



The impact of age uncertainty on the correlation of this record is apparent; even when correlated against itself, only 1.0% of the ensembles have r-values greater than 0.9, and the median correlation is 0.7. However, all of the correlations remain significant, even after accounting for autocorrelation, indicating that age uncertainty and the structure of the timeseries does not preclude the possibility of significant correlations.

Second, correlations are spectrally blind: they lump together all timescales, without regard for the dynamics that govern them. Because dynamical systems with large spatial scales tend to have long timescales as well, it is natural to expect that the GISP2 and Hulu records may vary in concert over millennial-scale events, but not necessarily shorter ones. Lastly, since age uncertainties will disproportionately affect high-frequency oscillations (e.g. Comboul et al., 2014), they would obfuscate such correlations even in perfectly synchronous records.

If the goal is to align features of interest, one solution would be to extract such features via Singular Spectrum Analysis (Vautard and Ghil, 1989; Vautard et al., 1992), then correlate such features only using the tools above. Yet another solution would be to use dedicated methods from the timeseries alignment (e.g. dynamic time warping) literature.

Generally, age uncertainties obscure relationships between records, while in rare cases creating the appearance of spurious correlations. It is appropriate to think of the ensemble correlation results produced by GeoChronR as a first-order estimate

of the age-uncertain correlation characteristics between timeseries, rather than a binary answer to the question "Are these two datasets significantly correlated?". However, as a rule of thumb, if more than half of the ensemble correlation results are significant, it is reasonable to characterize that correlation as robust to age uncertainty.

## 5.3   Age-uncertain calibration

A natural extension of ensemble correlation is ensemble regression, for which a common use case is calibrating a paleoenviron-

mental proxy "in time" by regressing it against an instrumental series using a period of overlapping measurements (Grosjean et al., 2009). We illustrate this by reproducing the results of Boldt et al. (2015), who calibrated a spectral reflectance measure of chlorophyll abundance, relative absorption band depth (RABD), to instrumental temperature in Northern Alaska. For each iteration in the analysis, a random age ensemble member is chosen and used to bin the RABD data onto a 3-year interval. The instrumental temperature data, here taken from the nearest grid cell of the GISTEMP reanalysis product (Hansen et al.,

2010), are also binned onto the same timescale, insuring temporal alignment between the two series. GeoChronR then derives an ordinary linear regression model, and then uses that model to "predict" temperature values from 3-year-binned RABD data back in time. This approach propagates the age uncertainties both through the regression (model fitting) and prediction process.

The function `plotRegressEns` produces multiple plots that visualize the key results of age-uncertain regression, and additionally creates an overview "dashboard" that showcases the key results (figure 6). The first row of figure 6 illustrates

the impact of age uncertainty on the regression modelling. In this example, the distribution of the modeled parameters, the slope and intercept of the regression equation, show pronounced modes of their distributions near $150°C^{-1}$ and $-130°C$, respectively, but with pronounced tails that include models with much lower slopes. This is also apparent in the scatterplot in the central panel of the top row, which illustrates the distribution of modeled relationships. Although the tendency for robust relationships are clear, models with slopes near zero also occur, suggesting that in this use case, age uncertainty can effectively




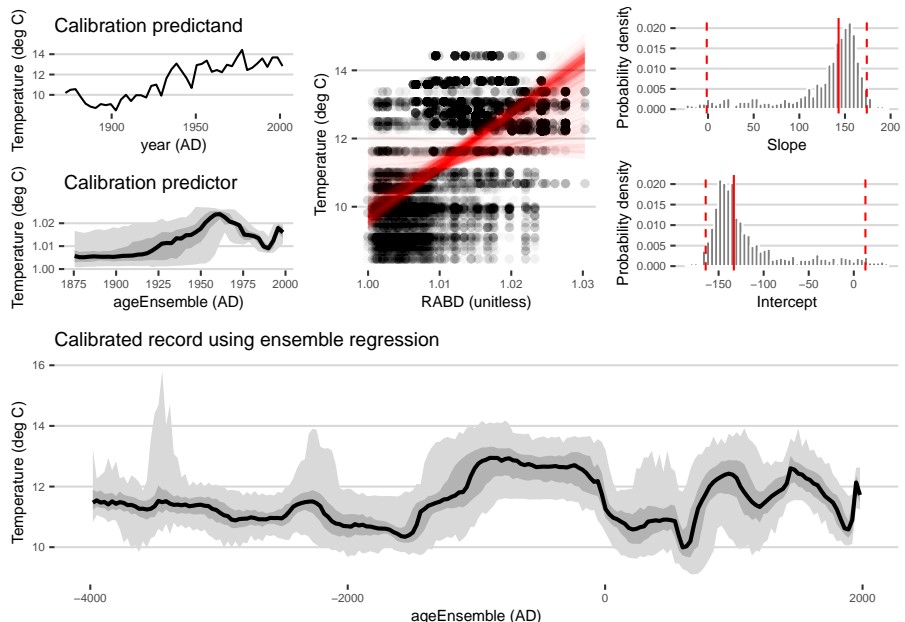

**Figure 6.** Results of ensemble regression. Top row, left: the calibration-in-time predictor and predictand. The predictand shows the effect of age uncertainty, the median is shown in black, and the 50 and 95% highest probability density regions are shown in dark and light gray, respectively. Top row, middle: Scatterplot showing the relation between RABD and Temperature. Points from 100 age ensemble members plotted with transparency to highligh data overlap. Ensemble regression models plotted as semi-transparent red lines. Top row, right: distribution of regression model slope (top) and intercepts (bottom). Vertical red lines show the 50th (solid), 2.5th and 97.5 (dashed) percentiles. Bottom row: reconstructed temperature using age ensembles and regression model ensembles. The median estimate is shown in black, and the 50 and 95% highest probability density regions are shown in dark and light gray, respectively.

destroy the relationship with instrumental data. The impact of this variability in modeled parameters, as well as the effects of age uncertainty on the timing of the reconstruction, are shown in the bottom panel of figure 6. The results shown here are consistent with those presented by Boldt et al. (2015), and we refer readers to that study for a full discussion of the implications of their results.

5    We note that there are use cases where regressing one age-uncertain variable onto another is called for, and `regressEns` supports such applications as well.

### 5.4 Arctic spatiotemporal variability over the Common Era

The previous use cases have highlighted age-uncertain analyses at one or two locations. Yet quantifying the effects of age uncertainty can be even more impactful over larger collections of sites. Here we showcase how to use GeoChronR to perform
10    age-uncertain principal components analysis (PCA), also known as Monte Carlo Empirical Orthogonal Function (MCEOF) analysis, pioneered by Anchukaitis and Tierney (2013). When seeking to analyze a large collection datasets, the first, and often





most time-intensive, step is to track down, format, and standardize the data. Fortunately, the emergence of community-curated standardized data collections (e.g. PAGES2K Consortium, 2013; Emile-Geay et al., 2017; Kaufman et al., 2020b; Konecky et al., 2020) can greatly simplify this challenge. In this example, we examine the Arctic 2k database (McKay and Kaufman, 2014), and use GeoChronR and the LiPD Utilities to filter the data for temperature-sensitive data from the Atlantic Arctic with
age ensembles relevant to the past 2000 years.

   Once filtered, the data can be visualized using `plotTimeseriesStack`, which is an option to quickly plot all of the time-series, on their best-estimate age models, aligned on a common horizontal timescale (figure 7). Although all of the datasets are relevant to Arctic temperatures over the past 2000 years, they span different time intervals, with variable temporal resolution. It is also clear that there is a lot of variability represented within the data, but it is difficult to visually extract shared patterns
of variability. Ensemble PCA is well-suited to the modes of variability that explain the most variance within these data, while accounting for the impact of age uncertainty.

   As in correlation and regression, aligning the data onto a common timescale is required for ensemble PCA. All but two of these datasets are annually resolved, and the other two have 5-year resolution, so it is reasonable to average these data into 5 year bins. Furthermore, because many of the records do not include data before 1400 CE, we only analyze the period from
1400 to 2000 CE. The data are now prepared for the ensemble PCA calculation, following a few choices in methodology and parameters. Because the data analyzed here have variable units, and we are not interested in the magnitude of the variance (only the relative variability between the datasets), we choose to use a correlation, rather than covariance, matrix. Next, we choose the number of components to estimate. After the analysis, a scree plot is used to determine the number of significant components. We want to estimate several more components than we anticipate will be meaningful. For this use case, we estimate eight
components.

   We now conduct the ensemble PCA, including null hypothesis testing, for 100 ensemble members. For a final analysis, 1000 ensemble members is standard, however the analysis can be time consuming and 100 members is appropriate for exploratory analyses. First, we plot the ensemble variance explained results for the data and the null hypothesis as a scree plot (figure 8). This represents how the variance explained by each component declines with each mode, for both the data and the null
hypothesis. Due to age uncertainty, the resulting variance explained is a distribution, which we compare to the 95th quantile of the null hypothesis ensemble. Figure 8 indicates that the first two components are clearly distinguished from the null. The third component is borderline, with the variance explained by the median of the ensemble near the null. Therefore, we will focus our investigation on the first two modes.

   The spatial and temporal results of the first two principal components are shown in figure 9. The first PC is dominated by
consistently positive loadings across the North Atlantic, suggesting that this is a regionally persistent mode of variability, and indicating that none of the datasets are negatively correlated with this mode. The corresponding timeseries shows multidecadal variability, with values declining until the 18th century, before increasing into the 20th century. Based on the region-wide coherence of the loading pattern and the similarity of the timeseries to regional temperature reconstructions (Hanhijärvi et al., 2013; McKay and Kaufman, 2014; Werner et al., 2018), the first PC likely reflects the primary pattern of regional temperature
variability. Notably, the uncertainties in the PC1 timeseries, and the loadings in the spatial pattern, are generally small. This,



**Figure 7.** Temperature-sensitive timeseries from the Atlantic Arctic in McKay and Kaufman (2014), arranged vertically by variable. Each timeseries is shown on its own scale to the right or left of each timeseries.





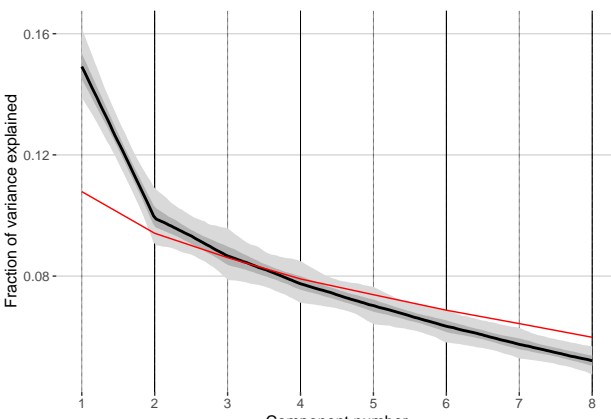

**Figure 8.** Ensemble PCA scree plot, showing the fraction of variance explained as a function of component (or eigenvalue) number. The ensemble PCA results for the data have uncertainty due to age-uncertainty, and themedian is shown in black, and the 50 and 95% highest probability density regions are shown in dark and light gray, respectively. The 95th percentile of the null hypothesis test is shown in red.

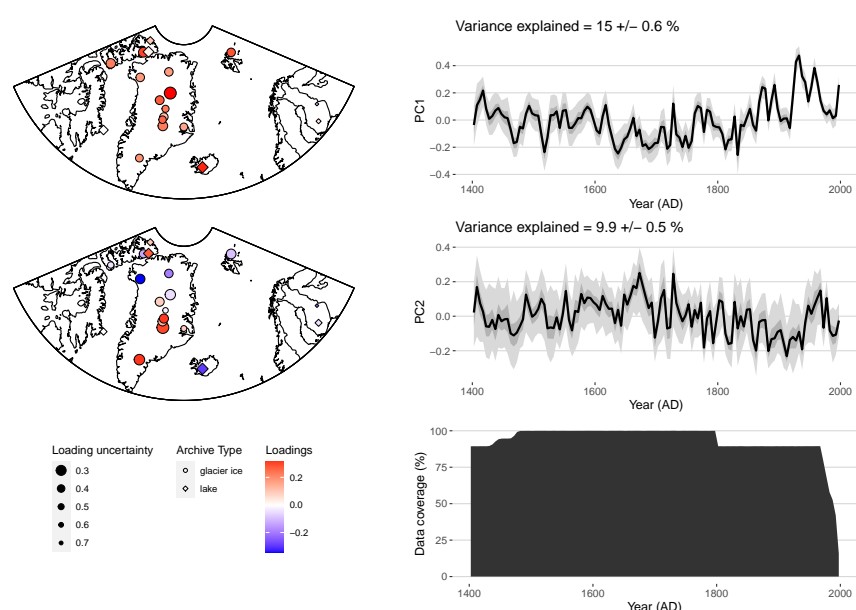

**Figure 9.** The spatial loading pattern (left) and timeseries (right) for the first principal component shown in the first row. The results for the second PC are shown in the second. The data density through time is shown in the bottom right corner. For the maps, the median loadings of the ensemble are shown by the color scale, and the standard deviation of the loadings across ensemble members is depicted by the size of the markers, with larger markers showing smaller uncertainties. For the timeseries plots, the median of the ensemble is shown in black, and the 50 and 95% highest probability density regions are shown in dark and light gray, respectively.





combined with the large amount of variance explained by PC1 relative to the null hypothesis (figure 8), suggests this is a significant mode of variability that is robust to age uncertainty. This makes intuitive sense, since the primary features of this pattern are century-long trends in temperature, a timescale that substantially exceeds the age uncertainty in these data (McKay and Kaufman, 2014).

The second PC shows considerable more variability in its spatial loading pattern, and a larger impact of age uncertainty. Generally, the loadings suggest a north-south dipole over Greenland for this mode, with positive loadings present in much of southern Greenland, with negative loadings in present in much of the northern part of the region. There is a much larger impact of age uncertainty on the loadings in PC2 than in PC1, illustrated by the size of the markers on the map, which are inversely related to the standard deviation of the loadings across the ensemble PCA results, such that smaller markers indicate larger

uncertainties. The PC2 timeseries includes more multidecadal variability than PC1 and is more impacted by age uncertainty. A key feature of the timeseries is a peak in values in the late 20th century, which occurs after the pronounced peak in PC1. This suggests that unlike the mid-20th century peak in warming apparent in most of the data, this later warming was dominated by contributions from southern Greenland, and counterbalanced by a decline in values in the northern Atlantic Arctic.

### 5.5   Orbital-scale variability in a deep-sea core

To illustrate the use of spectral analysis in GeoChronR, we consider a use case where the user seeks to identify the relative energy of oscillations at orbital (Milankovitch) periodicities in a deep-sea sediment core, and quantify the impact of age uncertainties on this assessment. Here we use a benthic paleotemperature record derived from the International Ocean Drilling Project core 846 (Mix et al., 1995; Shackleton, 1995), which covers the past 4.7 million years. For this assessment, we use an updated age model that was not generated within GeoChronR, rather, the age model was created via alignment to the benthic

$\delta^{18}$O stack of Lisiecki and Raymo (2005) using the HMM-Match algorithm (Lin et al., 2014; Khider et al., 2017). HMM-Match is a probabilistic method that generates an ensemble of 1000 possible age models compatible with the chronostratigraphic constraints; this ensemble was archived as a table in the associated LiPD file.

First we use `plotTimeseriesEnsRibbons` to visualize temperature, and the impact of age uncertainty, over the past 5 million years (figure 10).

This record displays three salient features:

- a long-term cooling trend characteristic of the late Neogene and Quaternary climate.

- quasi-periodic oscillations (the legendary Pleistocene Ice Ages)

- non-stationary behavior, related to the well-known mid-Pleistocene transition from a "41k world" to a "100k world" somewhere around 0.8 Ma (Paillard, 2001; Lisiecki and Raymo, 2005; Ahn et al., 2017).

For tractability, let us focus on the last million years, which cover the Quaternary Era. Over this interval, the time increments ($\Delta t$) are sharply peaked around 2.5 ka, spanning 0 to about 7.5 ka. From this point, there are two ways to proceed: 1) use methods that explicitly deal with unevenly-spaced data, or 2) interpolate to a regular grid and apply methods that assume even





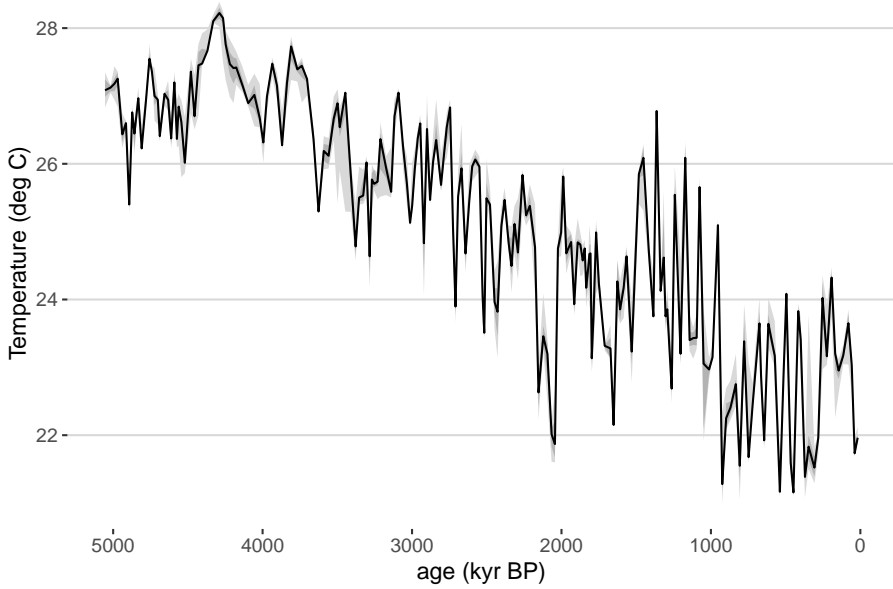

**Figure 10.** Temperature reconstruction from IODP 846. The median ensemble member is shown in black, with the 50 and 95% highest-density probability ranges shown in dark and light gray, respectively.

spacing (see section 3.4 ). Here we will use both approaches and highlight two of the four spectral methods implemented in GeoChronR: REDFIT and MTM.

We use the `computeSpectraEns` function to calculate the spectra for 1000 ensemble members using the REDFIT approach (figure 11). It is clear that the data contain significant energy (peaks) near, but not exactly at, the Milankovitch period-

5 icities (100, 41, 23, and 19 kyr). These periodicities, particularly those associated with eccentricity (100 kyr) and precession (23 and 19 kyr), rise above the null hypothesis (the 95% quantile from an autoregressive process of order one, see Mudelsee et al. (2009)). The obliquity periodicity is relatively weak, reaching just below the AR(1) benchmark.

The Lomb-Scargle periodogram used by REDFIT is a common way to deal with unevenly-spaced timeseries, but like all periodograms, it is inconsistent: the uncertainty about the spectral density at each frequency does not decrease with the number

of observations. This is mitigated somewhat with the application of Welch's Overlapping Segment Averaging, whose parameter choices (number of widows and degree of overlap) is not backed by theory. In contrast, MTM (Thomson, 1982) is an optimal estimator, which is consistent (the more observations, the better constrained the spectral density), and its parameter choice is explicit. Formally, MTM optimizes the classic bias-variance trade off inherent to all statistical inference. It does so by minimizing spectral leakage outside of a frequency band with half-bandwidth equal to $pf_R$, where $f_R = 1/(N\Delta t)$ is the

Rayleigh frequency, $\Delta t$ is the sampling interval, $N$ the number of measurements, and $p$ is the so-called *time-bandwidth product* (Ghil et al., 2002). $p$ can only take a finite number of values, all multiples of 1/2 between 2 and 4. A larger $p$ means lower variance (i.e. less uncertainty about the power), but broader peaks (i.e. a lower spectral resolution), synonymous with more





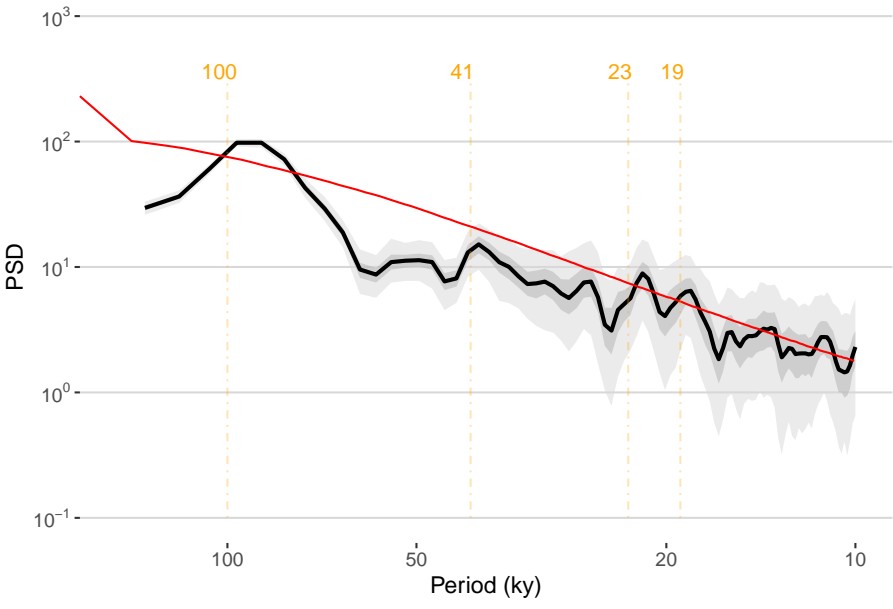

**Figure 11.** REDFIT spectrum for IODP 846. Median spectrum shown in black, with the 50 and 95% highest-density probability ranges shown in dark and light gray, respectively. Periodicities of interest (in kyr) shown in orange.

uncertainty about the exact location of the peak. So while MTM might not distinguish between closely spaced harmonics, it is much less likely to identify spurious peaks, especially at high frequencies. Several formal tests have been devised for both methods, allowing us to ascertain the significance of spectral peaks under reasonably broad assumptions. We show how to use MTM's "harmonic F-test" below.

However, classic MTM can only handle evenly-spaced data. Since the data are close to evenly-spaced, is is reasonable to interpolate them using standard methods. Both interpolation and MTM are implemented with the (astrochron Meyers, 2014) package, which GeoChronR employs.

To the spectral distribution itself we can add the periods identified as significant by MTM's F ratio test. GeoChronR estimates this by computing the fraction of ensemble members that exhibit a significant peak at each frequency. One simple criterion for gauging the level of support for such peaks given age uncertainties is to pick out those periodicities that are identified as

significant above a certain threshold (say, more than 50% of the time). For consistency with REDFIT, we define the null as an AR(1) process fit to the data, but GeoChronR supports two other nulls: a power-law null and a fit to the spectral background (Mann and Lees, 1996). Both follow the astrochron implementation.

Figure 12 shows a few differences between the REDFIT estimate (figure 11 and the MTM estimate. First, this ensemble of

spectra exhibits a clear power law behavior from periods of 5 to 100 ky, which in this log-log plotting convention manifests as a linear decrease. This is part of the well-documented "continuum of climate variability" (Huybers and Curry, 2006; Zhu





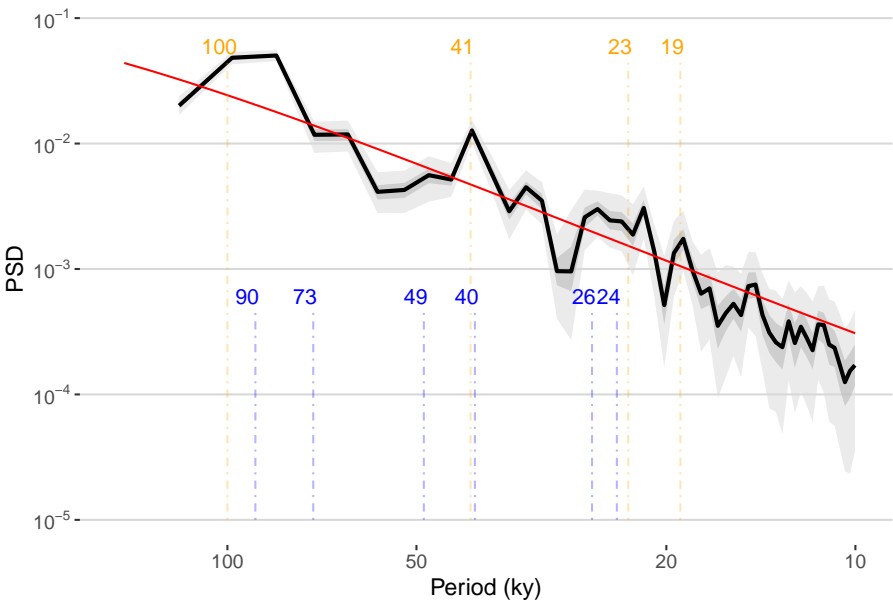

**Figure 12.** MTM spectrum for IODP 846. Median spectrum shown in black, with the 50 and 95% highest-density probability ranges shown in dark and light gray, respectively. Periodicities of interest (in kyr) shown in orange, and the periodicities identified as significant by the F ratio test are shown in blue.

et al., 2019), which is conspicuously absent from the Lomb-Scargle (REDFIT) estimate, known to be extremely biased in its estimate of the spectral background (Schulz and Mudelsee, 2002).

Secondly, the MTM version with this time-bandwidth product is sharper than REDFIT, with more well-defined peaks, particularly for the obliquity period (41 ky), which clearly exceeds the 95% confidence limit. Here it is helpful to take a step

back and contemplate our null hypothesis of AR(1) background, and the possibility that we might be underestimating the lag-1 autocorrelation, hence making the test too lenient. More importantly, the presence of scaling behavior (power law decrease) suggests that this should be a more appropriate null against which to test the emergence of spectral peaks. In GeoChronR, this can be done by specifying `mtm_null = "power_law"` in the function call.

Using a power law null hypothesis makes a few cycles appear non significant, but many remain (not shown). However,

carrying out a test simultaneously at many periodicities is bound to affect assessments of significance, via the multiple comparisons problem (Vaughan et al., 2011). In addition, sedimentary processes (and many processes in other proxy archives) tend to smooth out the signal over the depth axis, making comparisons at neighboring frequencies highly dependent (Meyers, 2012).

One solution is to use predictions made by a physical model about the frequency and relative amplitude of astronomical cycles (Meyers and Sageman, 2007). This approach, however, is not be applicable to all spectral detection problems. Ultimately,

the user must think deeply about the null hypothesis and the most sensible way to test it. Readers are invited to consider the





literature for a deeper exploration of these questions (e.g., Vaughan et al., 2011; Meyers, 2012, 2015; Meyers and Malinverno, 2018).

As with all statistical analyses in the paleosciences, there are no universal solutions or parameter choices. The approaches implemented in GeoChronR, especially with default choices, are best considered as exploratory tools. They are intended to
provide insight into the impacts of age uncertainty on power spectra, and to help users tailor their null hypotheses to their scientific questions.

## 6   Conclusions

GeoChronR provides user-friendly access to common age modeling tools in the paleosciences, along with intuitive visualization of the results. GeoChronR leverages the power of ensembles to propagate chronological uncertainties to subsequent stages of
paleoscientific analysis. While the approach does not address all aspects of uncertainty, it is quite general and provides key insights into which results are robust to chronological uncertainty, and which are not.

Although the focus has been on illustrating how to perform common tasks in a concise and intuitive way, GeoChronR also has the underlying infrastructure to support customized analyses for users seeking to address more complex questions (e.g. Thomas et al., 2018). At this stage, the paleoscience community is still only scratching the surface of age-uncertain analysis, so
many extensions are possible. We hope this article encourages the community to extend and expand this open-source package to achieve many more scientific goals than we could possibly enumerate here.

Looking forward, we suggest that the next major direction for age-uncertain analysis may not be technical, but philosophical in nature. Thus far, the community has focused on quantifying the range of possibilities presented by age uncertainty on a record-by-record basis, and GeoChronR has followed that approach. In the future, one could envision developing approaches
that leverage information from neighboring records. While idealized studies have laid the groundwork for using common forcings and/or covariance structures to do so (Werner and Tingley, 2015), much remains to be done to develop such a multi-site assessment of chronological uncertainties.

GeoChronR is open-source community-software, and has benefited substantially from multiple contributors and input from early adopters and workshop participants. We welcome feedback and strongly encourage contributions and enhancements, via
the GitHub issue tracker.

*Code availability.* All of the code used in GeoChronR is open and available at https://github.com/nickmckay/geochronr, and we welcome contributions and extensions to the package. The Rmarkdown code used to create this manuscript is available at https://github.com/nickmckay/geochronr-paper.

*Data availability.* All of the data used in this paper are publicly archived, and available as LiPD files at http://lipdverse.org/geochronr-
examples/.



*Competing interests.* The authors declare no competing interests.

*Disclaimer.* The software described here is provided under the MIT License, and is provided "as is", without warranty of any kind, express or implied, including but not limited to the warranties of merchantability, fitness for a particular purpose and noninfringement. In no event shall the authors or copyright holders be liable for any claim, damages or other liability, whether in an action of contract, tort or otherwise,

5   arising from, out of or in connection with the software or the use or other dealings in the software.

*Acknowledgements.* This project was funded by the US National Science Foundation Geoinformatics Program (EAR-1347221). We thank the participants of the 2016 and 2017 GeoChronR workshops for their contributions to the testing and design, which dramatically improved this package.



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
