# Peer review of "GeoChronR – an R package to model, analyze and visualize age-uncertain paleoscientific data"

_Geochronology, 2020_

## Short Comment (SC1) · 16 Sep 2020

Dear Authors,

First of all congratulations for putting together this R package which truly fills a niche in data analysis of sedimentary climate archives! Although, I am not a professional R user, I do know my way around the software and have worked with sedimentary climate proxy data from the data analysis side, and I would like to make a few comments on the usage of the package and its documentation in the MS to better your package and paper.

[Figure]

The MS describes the content of the R package nicely. It is easy to follow and understandable for researchers coming from various fields.

However, the usage of the package is not that well supported. In some cases, the help files in R are lacking complete-documentation and the user can only figure out the correct input when an error is shown. For example, in the computeSpectraEns {geoChronR} function, one does not know from the help file, the exact names of the possible inputs for the method argument. Of course the user can guess "Lomb-Scargle", " REDFIT", "MTM", but since R is case sensitive and only "mtm" is mentioned in the usage the user has to try out different options for "Redfit" let's say. While, when the error pops up: "Error in computeSpectraEns(adat[, 1], adat[, 2], max.ens = 1000, method = "REDFIT", : Unknown method: Valid choices are: 'mtm', 'redfit', 'nuspectral', or 'lomb-scargle'", it becomes clear what the correctly spelled options are for methods.

Staying in the same section. It is not documented exhaustively that which argument is used by which method. Yes, the topic is addressed in the MS (Sects. 3.4. and 5.5.) and one can read the original documentation of the e.g. dplR:REDFIT function, but either (i) this should be called to the users attention that you refer to the original documentation, or (ii) it should be briefly described in the package help.

In fine, I strongly suggest providing all-round tutorials for the examples presented in the MS, for example (i) on the Temperature reconstruction from IODP 846 shown in Fig. 10 of the MS, similarly detailed as in the SISAL_v2 paper (https://essd.copernicus.org/preprints/essd-2020-39/) for the OxCal modeling (https://zenodo.org/record/3586280), or (ii) for the PCA examples (Sect. 5.4) using a set of records directly from the iso2k database, which would promote that database even more.

I am fully aware that this is additional work, but I am convinced that such tutorials for each of the relevant sections of the MS would surely broaden the audience and more importantly, the user base of geoChronR.

[Figure]

Yours sincerely,

István Hatvani

---

## Referee Comment (RC1) · Anonymous Referee #1 · 27 Sep 2020

GeoChronR represents a nice new addition to the field of paleoclimatology and sediment-based paleoenvironmental reconstruction by presenting what is essentially a 'one-stop shop' for age uncertainty analysis in paleoenvironmental sequences, and includes code for analyzing data calibrated via a variety of different methods (e.g. layer counting, U-Th, radiocarbon), etc. This removes the need to use different software for each type of proxy archive, and as the authors note, should speed up the analysis of large ensembles of paleoclimatic data.

I think the paper is mainly fine as is, just some minor changes to highlight the flexibility of the package and hone some of the science examples presented would be good.

[Figure]

That being said, given that this system uses the LiPD file format, I think it would be useful for the authors to include explicit mention of other utilities that exist (e.g. in Python) for converting other file formats (e.g. text files from NOAA Paleoclimatology). Our community still has a lot of different 'standards' for data archiving floating around, so it would be nice to emphasize that this software package can be used on other data provided the authors are willing to convert files to the correct format.

I find the incorporation of correlation analysis especially compelling.

In the examples, for instance in 5.2, the purported lack of correlation between GISP2 and Hulu is quite controversial as I'm sure the authors know. I would like a bit more discussion of the reasons for this - I think the two records can be very strongly linked, but still not show strong Pearson correlations. GISP and sometimes Hulu seems to show 'on' or 'off' values - e.g. the record is jumpy, akin to a Dirac delta function especially for some of the rapid millennial-scale Heinrich events. Pearson product moment correlation, even if spectrally filtered, might not be appropriate, since it focuses on linear association. Would a ranked correlation metric like Spearman's rho show different results? Incorporating alternate metrics of correlation into GeoChronR is in my opinion not needed for this release, but it could be useful to mention here, along with the suggestion that users can use the age model output to build their own analyses using alternate correlation metrics, different assumptions about the underlying distributions of the data. I imagine that is the hope anyways with such a flexible package.

I may be wrong, but haven't the Greenland ice core chronologies been revised to create chronologies that produce a stronger relationship between greenland oxygen isotopes and the Hulu record?

I really like the analysis with the Arctic temperature database and hope a more substantial paper on the meaning of these PCs is forthcoming. I wonder if PC2 would be more easily interpreted if varimax rotation or some method were applied.

---

## Author Comment (AC1) · 7 Oct 2020

We thank the reviewer for their time reading our manuscript and trying out the package, as well as for their suggestion about improving function documentation and tutorials. We completely agree with this suggestion, indeed, since submission of the manuscript we have been working to build out and streamline our documentation, including full-fledged tutorials.

Although we will continue to add to the documentation and articles, a complete documentation website is now available at https://nickmckay.github.io/GeoChronR/ . This includes (at present) nine tutorials, as well as documentation for every function in the

package.

Thank you for catching the lacking documentation in computeSpectraEns(). We have updated and corrected that oversight, you can see the updated reference page here: https://nickmckay.github.io/GeoChronR/reference/computeSpectraEns.html .

We hope that you will continue to use, comment on, and make suggestions for how we can continue to improve geoChronR. We will also add a section in our documentation concerning community engagement in improving upon the package. This includes raising issues on GitHub for bugs and improved documentation as well as direct contributions to the core package and associated documentation through pull requests.

Nick McKay, Julien Emile-Geay and Deborah Khider

---

## Referee Comment (RC2) · Anonymous Referee #2 · 11 Oct 2020

This ms is clearly written and makes a convincing case for standardizing paleo data for subsequent compilation studies. It shows a range of very helpful developments in compiling and compiling multiple records, enabling much more informed analyses than currently available. It is high time for the paleo-community to stop neglecting chronological uncertainties. I hope that this initiative will be taken up and used to make more robust paleoclimatic inferences.

Pending just the addition of a few references, I suggest accepting this manuscript for publication.

On p2, please also cite some earlier papers on comparisons between records

that take into account chronological uncertainties: doi:10.1177/0959683607075857 doi:10.1016/j.quascirev.2008.07.009 doi:10.1016/j.quascirev.2010.11.012

For the comparison of Hulu and GISP2, it might also be useful to cite doi:10.1002/jqs.1330 which calculates 'event probabilities' within time-windows for individual records, and then calculates probabilities for synchronous reactions between two records as the product of these probabilities for each time window.

For section 5.4, a citation to a recent compilation paper in Science might also be helpful here: doi:10.1126/science.aay5538.

Details p16 line 5, significant caption Fig. 8: the median

---

## Short Comment (SC2) · 17 Oct 2020

Having myself worked for over thirty years in the scientific area targeted by this interesting paper – age-uncertain paleoscientific data analysis – it is my pleasure and privilege to add some remarks. Geochronology Discussions is one of the EGU's family of discussion journals, which are important to have since they allow our students to get a more intimate understanding of how the science of the paleoclimate and other disciplines evolves, how we paleoclimate researchers struggle for the truth. Inevitably, my contribution is biased towards own work, in particular my book on climate time series analysis (Mudelsee, 2014), to which for the sake of brevity here I sometimes refer to

as "Book", and of which – this is for those people interested in the history and also in who wrote first on a subject – the first edition appeared in 2010.

I find the paper well written, easy to read and understand, despite (because of?) the fact that nearly no equations are given. It shows that still one can speak in a rational manner about quantitative methods of data analysis. What I like most is the emphasis the paper puts on the usefulness of having available, in addition to a timescale, $\{t(i)\}_{i=1}^{n}$, where $t$ is time or age, $i$ is a counter, and $n$ the sample size, also a set of simulated timescales $\{t^*(i)\}_{i=1}^{n}$. This set of simulation results is useful to have for the determination of the *full* errors of statistical estimations on paleoclimatic time series. That means, we have to take into account not only (1) measurement error of a climate proxy variable, $\{x(i)\}_{i=1}^{n}$, and (2) proxy error (which usually exceeds the measurement error by far), but also (3) timescale errors. The software GeoChronR presented by McKay et al. (2020) serves well those meticulous researchers embarking on a full error determination, since GeoChronR (1) comprises several timescale construction algorithms and (2) is written in the R language, which is gaining much popularity now in geosciences.

However, the meticulous researcher should be informed that the supply of $\{t^*(i)\}_{i=1}^{n}$ by GeoChronR is just the first of two milestones she or he has to reach. The second milestone – the development and utilization of statistical methods for processing $\{t^*(i)\}_{i=1}^{n}$ – this is in my view the major one. It has to be mentioned that, unfortunately, method development and theoretical derivations for uncertain timescales has never been high on the agenda of statistical science – at least this is my impression from the study of the literature in this science and from the news for members of the Royal Statistical Society (to which I belonged for about 20 years). In what follows I will mention some statistical estimation methods, which are described by McKay et al. (2020: Section 3 therein). It is the privilege of an external commenter in this discussion journal to not having to perform an exhaustive review but be allowed to pick out what is deemed interesting. I may add that I fail to find exhaustive the other comments in the interactive

discussion (date of writing, 17 October 2020).

**1  Correlation**

A major problem imposed by uncertain ages occurs only if the timescales for $X$ and $Y$ are not completely dependent. A minor problem for completely dependent timescales may be that autocorrelation estimates, such as the persistence time for an AR(1) process on an unevenly spaced time grid (Mudelsee, 2002), may be influenced – but this can be safely ignored since this affects only block length selection in the bootstrap resampling for uncertainty determination, and block length selection is not very influential here (Mudelsee, 2014: Table 7.2 therein). A practical example for completely dependent (although uncertain) timescales is a marine sediment core, where on sample material from identical depths proxy measurements are done, such as oxygen ($X$) and carbon ($Y$) isotopic compositions.

For the case of not completely dependent timescales, McKay et al. (2020) present the binning approach. It should be mentioned that this works only in the presence of autocorrelation in the data $(X, Y)$ generating process because then also time-distant points may "know" to some degree about the current point. In my Book (Mudelsee, 2014: Section 7.5 therein), I give bin width selection rules based on the persistence time estimates (for $X$ and $Y$) and I show Monte Carlo simulation results obtained on artificial data in order to test the method. One results is that binning outperforms interpolation. A recent implementation of "my" binning approach for correlation estimation is the R software BINCOR (Polanco-Martinez et al., 2019).

As regards the assessment of the significance of the correlation coefficient (which, after all, is an *estimate* of the true but unknown population correlation coefficient), McKay et al. (2020) are correct in their assessment that the standard way is via a statistical test (of the null "population correlation coefficient equals zero") that assumes

normal shapes (for $X$ and $Y$) and serial independences (of $X$ and $Y$) – unfortunately! What the Monte Carlo experiments on correlation estimation, summarized in the Book (Mudelsee, 2014: Section 7.3 therein), teach us is sobering.

1. A confidence interval as uncertainty measure is superior to a $P$-value of a significance test because it bears more quantitative information. It allows you to compare two correlations, whether or not one association is stronger than another. A practical example is when you wish to construct a rank list of surface-air temperature measurement stations in Europe in terms of the strength of the correlation with the North Atlantic Oscillation.

2. Serial dependence can be taken into account by pairwise block bootstrap resampling (Mudelsee, 2014: p. 279 therein). The resampling preserves the distributional shapes, and the pairwise manner preserves (over the block length) the serial dependence or autocorrelation structures. Serial dependence can also be taken into account in a "classical" manner (i.e., using formulas instead of employing resampling or simulation approaches) via the effective number of degrees of freedom, as mentioned by McKay et al. (2020: Section 3.1 therein) as the first approach of GeoChronR.

3. The presence of non-Gaussian shapes *completely* invalidates determined classical uncertainty measures. Then you cannot trust at all a $P$-value or a confidence interval obtained in a classical manner. (This is what you get if you press the button in standard "office software".) The only remedy that yields acceptably accurate uncertainty measures is pairwise block bootstrap resampling enhanced by computing-intensive calibration (Mudelsee, 2014: Chapter 7 therein). There is a Fortran 90 software on this that employs parallel computing (Ólafsdóttir and Mudelsee, 2014).

4. Spearman's (1904, 1906) rank correlation coefficient is more robust (i.e., it delivers more accurate uncertainty measures in the presence of violations of made

distributional assumptions) than Pearson's (1896) coefficient. (By the way, it is interesting from a philosophy-of-science viewpoint to see how Pearson reacted when he became aware of Spearman's papers.)

**2 Regression**

First, the observation by McKay et al. (2020: Section 3.2 therein) that age-uncertainties plague also the calibration of proxy variables is important. GeoChronR's error propagation into a set of calibration curves is useful. However, a problem with proxy calibration is that here both variables (the proxy and the indicated climate variable) do show measurement uncertainties – and this leads in ordinary least squares regression (as done by GeoChronR) to a down-biased slope estimate. What is needed is a bias correction, and the Book (Mudelsee, 2014: Chapter 8 therein) gives two ways to perform this.

As regards regression, one may also mention trend estimation, for example, change-point detection on time series (Mudelsee, 2000; Mudelsee, 2009). Also this estimation target becomes noisier in the presence of timescale errors. However, using a hybrid resampling approach (block bootstrap for obtaining $\{x^*(i)\}_{i=1}^n$, parametric for $\{t^*(i)\}_{i=1}^n$), for which GeoChronR can be utilized, reliable uncertainty determination can be achieved, as Monte Carlo simulations demonstrate (Mudelsee, 2014: Chapter 4 therein). One may further mention nonparametric regression, where this hybrid approach also works; as an example, the paper by Mudelsee et al. (2012) is entitled "Effects of dating errors on nonparametric trend analyses of speleothem time series."

**3  Spectral Analysis**

The estimation of the true (but unknown) spectrum of the random component in a climate-data generating process is important for reasons also McKay et al. (2020) give: you can study peaks versus background noise; external drivers; leads and lags; and so forth – a spectrum allows you to learn about the physics of the system. Typical records from paleoclimate archives are unevenly spaced since the accumulation process of an archive usually is not constant over time and constant depths are sampled because of material requirements for the measurements. The various timescale construction algorithms, implemented in GeoChronR or elsewhere, can be used to study accumulation in detail. Even in the "untypical" situation of even spacing, it is then the age-uncertainty that introduces uneven spacing into the $\{t^*(i)\}_{i=1}^n$. In any way, spectral estimation for paleoclimate time series has to deal with uneven temporal spacing.

In my view, the Lomb-Scargle method is superior to other spectrum estimation methods since it is regression-based and therefore can be directly applied to unevenly spaced series. However, the raw periodogram (GeoChronR's first spectral approach) is bad to employ since it renders estimates with 100% relative error and also is an inconsistent estimator (i.e., the estimation error does not decrease with $n$), as we know for decades, see for example the Book (Chapter 5 therein) and the references cited therein. The periodogram therefore has to be combined with a segmenting approach, which trades spectral resolution for reduced estimation error. This advantage, not mentioned by McKay et al. (2020), combined with a test against AR(1) red noise, is the reason of the success of the REDFIT implementation (Schulz and Mudelsee, 2002); which constitutes GeoChronR's second spectral approach. It may be noted that there exists a version called REDFIT-X for cross-spectral analysis (Ólafsdóttir et al., 2016). For the purpose of studying timescale error effects on spectral estimates, I made experiments with adaptations of the REDFIT code (Mudelsee et al., 2009; Mudelsee, 2014: Section 5.2.8 therein) to quantify (1) how much wider spectral peaks get and (2) how much

higher the red-noise upper pertenciles get, but there is not yet a publishable code on this.

The other two spectral approaches mentioned by McKay et al. (2020) are wavelet decomposition and Thomson's (1982) multitaper. To the best of my knowledge, for both approaches there is (yet) no reliable implementation or code that is directly applicable to uneven temporal spacing. Likely it is this situation that led McKay et al. (2020: p. 11, line 8–9 therein) to advocate and put into GeoChronR the data pretreatment of "efficient linear interpolation". As many authors before me or Michael Schulz have demonstrated (e.g., Horowitz, 1974), and what I also regularly teach in my courses on climate time series analysis – interpolation means a step away from the original data. It is a dangerous activity. It introduces autocorrelation where there has been none before. If researchers are not trained, then some of them may be even entrapped to employ interpolation in order to boost up the sizes of the samples. This leads to too small uncertainty measures of estimations, to overstatements about the climate system, it damages the credibility of climate research. On top of that, there do exist various interpolation methods (linear, cubic spline, Akima spline, etc.). Which interpolation method to use? Just the "most efficient one"? – Dangerous. In my view the *only* way of having interpolation (to even spacing) in the arsenal of methods is if a meticulous researcher does embark on coding and performing extensive Monte Carlo simulations. In such experiments, the true properties of a data generating process can be prescribed. The properties (e.g., spectrum, $n$, spacing, noise level) should be close to what data you have, what the prior knowledge about the data indicates or what the geological–physical intuition tells you, the researcher. Then the distorting effects of the various interpolation methods can be quantified and compared. And only if the distorting effects can be shown as negligible, then one may safely proceed with interpolation and present results that are robust in the original statistical sense.

**4  Outlook**

It is great to see that the engagement of professional statisticians in paleoclimate research is growing. Certainly they can contribute to the issue of the statistical analysis of age-uncertain paleoscientific data.

However, it appears that the various climatological communities, who partly tend to associate themselves with the employed type of paleoclimate archive, are developing a tendency towards using timescale construction algorithms that have been designed within their own community: Oxcal (Ramsey, 2008) or BChron (Haslett and Parnell, 2008) for the radiocarbon community; Wheatley et al.'s (2012) method or BAM (Comboul et al., 2014) for the layered archives such as corals or ice cores; StalAge (Scholz and Hoffmann, 2011), iscam (Fohlmeister, 2012) or MOD-AGE (Hercman and Pawlak, 2012) for the speleothem community. This tendency is unhealthy for scientific progress because it reduces the open exchange among researchers about the optimal way of timescale construction.

Certainly specific archives have their own peculiarities (e.g., about whether hiatuses may occur, whether there exists prior geological-physical knowledge about minimum or maximum accumulation rates) – however, the situation could be improved if the various methods are allowed to be compared against each other (in terms of bias, standard error, etc.) within a Monte Carlo experiment. Scholz et al. (2012) have already done a comparison of methods, but a step beyond that work would be achieved if also the delivered uncertainty bands (around the age–depth curve) could be tested. For example, whether or not a 95% confidence interval for an age at a certain depth does indeed include the true (and known, since it is prescribed) age in 95% of the simulations (within simulation noise). We paleoclimate researchers, as all applied researchers, should apply robust tools that yield reliable estimations and reliable error bars also in the presence of (1) non-Gaussian distributions, (2) autocorrelation and (3) age-uncertainties. I am optimistic that the paper by McKay et al. (2020) and the supplied GeoChronR tool

can help to achieve this "Timescale Monte Carlo Comparison Experiment".

**GChronD**

Interactive
comment

**References**

Comboul, M., Emile-Geay, J., Evans, M. N., Mirnateghi, N., Cobb, K. M., and Thompson, D. M.: A probabilistic model of chronological errors in layer-counted climate proxies: Applications to annually banded coral archives, Clim. Past, 10, 825–841, 2014.

Fohlmeister, J.: A statistical approach to construct composite climate records of dated archives, Quat. Geochronol., 14, 48–56, 2012.

Haslett, J. and Parnell, A.: A simple monotone process with application to radiocarbon-dated depth chronologies, Appl. Stat., 57, 399–418, 2008.

Hercman, H. and Pawlak, J.: MOD-AGE: An age–depth model construction algorithm, Quat. Geochronol., 12, 1–10, 2012.

Horowitz, L. L.: The effects of spline interpolation on power spectral density, IEEE Transactions on Acoustics, Speech, and Signal Processing, ASSP-22, 22–27, 1974.

McKay, N., Emile-Geay, J., and Khider, D.: GeoChronR – an R package to model, analyze and visualize age-uncertain paleoscientific data, Geochronol. Discuss., https://doi.org/10.5194/gchron-2020-25, in review, 2020.

Mudelsee, M.: Ramp function regression: A tool for quantifying climate transitions, Comput. Geosci., 26, 293–307, 2000.

Mudelsee, M.: TAUEST: A computer program for estimating persistence in unevenly spaced weather/climate time series, Comput. Geosci., 28, 69–72, 2002.

Mudelsee, M.: Break function regression: A tool for quantifying trend changes in climate time series, Eur. Phys. J.-Spec. Top., 174, 49–63, 2009.
Mudelsee, M., 2014. Climate Time Series Analysis: Classical Statistical and Bootstrap Methods, 2nd edn. Springer, Cham, Switzerland, 454pp.

Mudelsee, M., Scholz, D., Röthlisberger, R., Fleitmann, D., Mangini, A., and Wolff, E. W.: Climate spectrum estimation in the presence of timescale errors, Nonlinear Process. Geophys., 16, 43–56, 2009.

Mudelsee, M., Fohlmeister, J., and Scholz, D.: Effects of dating errors on nonparametric trend analyses of speleothem time series, Clim. Past, 8, 1637–1648, 2012.

Ólafsdóttir, K. B. and Mudelsee, M.: More accurate, calibrated bootstrap confidence intervals for estimating the correlation between two time series, Mathematical Geosciences, 46, 411–427, 2014.

Ólafsdóttir, K. B., Schulz, M., and Mudelsee, M.: REDFIT-X: Cross-spectral analysis of unevenly spaced paleoclimate time series, Comput. Geosci., 91, 11–18, 2016.

Pearson, K.: Mathematical contributions to the theory of evolution—III. Regression, heredity, and panmixia, Phil. Trans. R. Soc. Lond. A, 187, 253–318, 1896.

Polanco-Martinez, J. M., Medina-Elizalde, M. A., Sanchez Goni, M. F., and Mudelsee, M.: BINCOR: An R package for estimating the correlation between two unevenly spaced time series, The R Journal, 11, 170–184, 2019.

Ramsey, C. B.: Deposition models for chronological records, Quat. Sci. Rev., 27, 42–60, 2008.

Scholz, D. and Hoffmann, D. L.: StalAge – an algorithm designed for construction of speleothem age models, Quat. Geochronol., 6, 369–382, 2011.

Scholz, D., Hoffmann, D. L., Hellstrom, J., and Ramsey, C. B.: A comparison of different methods for speleothem age modelling, Quat. Geochronol., 14, 94–104, 2012.

Schulz, M. and Mudelsee, M.: REDFIT: Estimating red-noise spectra directly from unevenly spaced paleoclimatic time series, Comput. Geosci., 28, 421–426, 2002.

Spearman, C.: The proof and measurement of association between two things, Am. J. Psychol., 15, 72–101, 1904.

Spearman, C.: 'Footrule' for measuring correlation, British Journal of Psychology, 2, 89–108, 1906.

Thomson, D. J.: Spectrum estimation and harmonic analysis, Proc. IEEE, 70, 1055–1096, 1982.

Wheatley, J. J., Blackwell, P. G., Abram, N. J., McConnell, J. R., Thomas, E. R., and Wolff, E. W.: Automated ice-core layer-counting with strong univariate signals, Clim. Past, 8, 1869–1879, 2012.

---

## Author Comment (AC2) · 16 Nov 2020

**GeoChronR response to SC2**

Monday, November 16, 2020

We thank Dr Mudelsee for offering his valuable perspective on this topic, and spending the time to share these comments. We take up his points in turn:

**On uncertainty**

We wholeheartedly agree that chronological uncertainties are only one of many types of uncertainty affecting paleoenvironmental timeseries. Indeed, we practice this every day in our own research. We agree that it is important to place chronological uncertainties in this broader context, and will do so in the revised paper.

**On correlations**

We thank Dr Mudelsee for bringing BINCOR [Polanco-Martinez et al., 2019] to our attention, as we were not aware of it. It seems like a useful and complementary way of going about the problem, and we will investigate incorporating it and its concepts in geoChronR in upcoming releases. The reason why this cannot be done quickly is that it makes several deep assumptions that may or may not be consistent with others that we make, and we need to think this over carefully.

We agree wholeheartedly with points 1, 2, 3, and 4. A confidence interval may be derived from the histogram of correlations, but we will work to export its summary as a a 95% CI, say. We agree that non-gaussianity can be an issue, which is why geoChronR's default behavior is to transform input data to a standard normal (via quantile mapping [van Albada and Robinson, 2007; Emile-Geay and Tingley, 2016]). Lastly, it is true that we could make it easier for users to use other formulations of the correlation coefficient, like Spearman's. We will incorporate rank-based correlation methods into the package on revision of the manuscript. The impact on significance estimates will be the same for the non-parametric tests (isopersistent, isospectral) implemented in geoChronR, so we expect this to be an easy transition.

[Figure]

**Fig. 1.** Benchmarling the Lomb-Scargle implementations in `lomb::lsp()` (orange curve), REDFIT (via the `dplR` package, [Bunn, 2008, green curve]) and `SciPy` [Virtanen et al., 2020, blue curve], as implemented in Khider et al. [2018]. The sample signal is a 20-year sinusoid sampled yearly over 1000 years.

**On regression**

We are aware that OLS is a biased estimator in the presence of timescale errors. One can imagine extensions like truncated total least squares [Van Huffel and Vandewalle, 1991; Van Huffel, 2004; Markovsky et al., 2010], or the ones suggested in "the book" [Mudelsee, 2013]. We will flag in the revised version of the paper that timescale uncertainties strongly interact with other uncertainties here, in a way that needs to be more rigorously assessed.

**On spectral analysis**

We disagree with the statement that "the Lomb-Scargle (LSP) method is superior to other spectrum estimation methods since it is regression-based and therefore can be directly applied to unevenly spaced series."

LSP is appropriate in many circumstances VanderPlas [2018], and we've used it in our own work [e.g., Khider et al., 2014], but there are many applications where it can be problematic. First, let us note that the weighted wavelet Z transform [WWZ Foster, 1996; Kirchner and Neal, 2013; Zhu et al., 2019] shares many of these characteristics, and performs very similarly on analytical benchmarks [Khider et al.]. However, there is an important difference between the implementation of the Lomb-Scargle algorithm in R (as used in GeoChronR) and Python (specifically, the SciPy package). This comparison was carried out on a simple, 20-year harmonic in Khider and Emile-Geay [2020], and is summarized in Fig. 1.

For such a simple and abundantly sampled harmonic signal, any good estimator should return something close to a delta function peaked at the $f_0$ frequency. We see that the SciPy implementation of LS achieves this, but the

the R and REDFIT implementations are extremely noisy, detecting many spurious peaks at high frequencies. Indeed, the signal to noise ratio is 4 to 15 orders of magnitude smaller in those implementations than in SciPy's, though REDFIT's spectrum is (by virtue of averaging) quite a bit smoother than the standard `lomb` implementation. This is quite a substantial difference, for which we cannot find an easy explanation. Indeed, the algorithm is the same, only the numerical implementation differs. In the revised paper, geoChronR users will be alerted to this important limitation.

We do agree that interpolation has serious downsides, but disagree with the blanket statement that it is always "dangerous". Indeed, in our tests, Lomb-Scargle detects many spurious features. The example from section 5.5 is one where the spacing is nearly equal, so the effects of interpolation are minimal. On benchmarks using synthetic data, the multitaper method (MTM) is far superior to LSP with WOSA on evenly-spaced series, and interpolation – used sparingly – provides a way to access these important features of MTM. Thus, we do not agree with a blanket condemnation of interpolation, though we agree that it must be used very carefully to avoid raising the sample size to spuriously high levels; in our practice, typically err on the side of coarse-graining the time-series to avoid this effect, and only use linear interpolation to avoid introducing spurious oscillations. We do agree that sanity checks of robustness are essential.

**On age modeling**

We agree with Dr Mudelsee that the proliferation of chronology modeling methods is problematic. This is why geoChronR only considers methods based on explicit statistical models, so that the assumptions may be examined on their scientific merits. Unfortunately, this is not the case for many of the other methods you bring up (e.g. StalAge), and we thus refrain from using those in the package or elsewhere.

Age model intercomparison would indeed an important application of `geoChronR`, as it provides a standardized platform upon which methods can be readily tested and compared. Although a thorough intercomparison is beyond the scope of this article, we very much hope to pursue it in future investigations, or to facilitate this task for other investigators. We will revise our discussion to mention recent work on this topic. While Scholz et al. [2012] have indeed performed a comparison of some methods, Parnell et al. [2011] had published on some of the same methods the year before, coming to somewhat different conclusions as Scholz et al. [2012]. More recently, Trachsel and Telford [2016] compared several of the methods included in geoChronR (including OxCal, BChron, and Bacon) on a reference, varved lake chronology. They conclude that "All methods produce mean age–depth models that are close to the true varve age, but the uncertainty estimation differs considerably among models." In particular, BChron is found to overestimate uncertainties in this context. There is thus plenty more to be done to document, benchmark and understand the effects of these various modeling choices, and our revised paper will point to the existing

work.

In regards to *"whether or not a 95% confidence interval for an age at a certain depth does indeed include the true (and known, since it is prescribed) age in 95% of the simulations"* this is called **coverage rate** in the statistical literature, and is indeed a property that paleoscientists should try to constrain for the various methods.

*"I am optimistic that the paper by McKay et al. (2020) and the supplied GeoChronR tool can help to achieve this "Timescale Monte Carlo Comparison Experiment"."* We share your optimism and thank you for these valuable comments, which helped improve the package and the paper.

**References**

Bunn, A. G.: A dendrochronology program library in R (dplR), Dendrochronologia, 26, 115–124, https://doi.org/10.1016/j.dendro.2008.01.002, 2008.

Emile-Geay, J. and Tingley, M.: Inferring climate variability from nonlinear proxies: application to palaeo-ENSO studies, Climate of the Past, 12, 31–50, https://doi.org/10.5194/cp-12-31-2016, URL `http://www.clim-past.net/12/31/2016/`, 2016.

Foster, G.: Wavelets for period analysis of unevenly sampled time series, Astron. Jour., 112, 1709, https://doi.org/10.1086/118137, 1996.

Khider, D. and Emile-Geay, J.: Benchmarking the Lomb-Scargle implementation in GeochronR, URL `https://github.com/KnowledgeCaptureAndDiscovery/autoTS/blob/master/notebooks/Methods/Lomb-Scargle%20Performance_RvsPython.ipynb`, 2020.

Khider, D., Athreya, P., and Emile-Geay, J.: Estimation of spectral peaks and continuum – a benchmark, URL `\url{https://github.com/KnowledgeCaptureAndDiscovery/autoTS/blob/master/notebooks/Methods/lomb_scargle_vs_wwz_analytical_benchmarks.ipynb}`.

Khider, D., Jackson, C. S., and Stott, L. D.: Assessing millennial-scale variability during the Holocene: A perspective from the western tropical Pacific, Paleoceanography, 29, 143–159, https://doi.org/10.1002/2013PA002534, URL `http://dx.doi.org/10.1002/2013PA002534`, 2014.

Khider, D., Zhu, F., Hu, J., and Emile-Geay, J.: LinkedEarth/Pyleoclim_util: Pyleoclim release v0.4.0, https://doi.org/10.5281/zenodo.1205662, URL `https://doi.org/10.5281/zenodo.1205662`, 2018.

Kirchner, J. W. and Neal, C.: Universal fractal scaling in stream chemistry and its implications for solute transport and water quality trend detection, Proceedings of the National Academy of Sciences, 110, 12 213–12 218, https://doi.org/10.1073/pnas.1304328110, 2013.

Markovsky, I., Sima, D. M., and Van Huffel, S.: Total least squares methods, Wiley Interdisciplinary Reviews: Computational Statistics, 2, 212–217, https://doi.org/10.1002/wics.65, 2010.

Mudelsee, M.: Climate time series analysis: Classical Statistical and Bootstrap Methods, Springer, 2nd edn., 2013.

Parnell, A. C., Buck, C. E., and Doan, T. K.: A review of statistical chronology models for high-resolution, proxy-based Holocene palaeoenvironmental reconstruction, Quaternary Science Reviews, 30, 2948 – 2960, https://doi.org/10.1016/j.quascirev.2011.07.024, URL `http://www.sciencedirect.com/science/article/pii/S0277379111002356`, 2011.

Polanco-Martinez, J. M., Medina-Elizalde, M. A., Goni, M. F. S., and Mudelsee, M.: BINCOR: An R package for Estimating the Correlation between Two Unevenly Spaced Time Series, The R Journal, 11, 170–184, https://doi.org/10.32614/RJ-2019-035, URL `https://doi.org/10.32614/RJ-2019-035`, 2019.

Scholz, D., Hoffmann, D. L., Hellstrom, J., and Bronk Ramsey, C.: A comparison of different methods for speleothem age modelling, Quaternary Geochronology, 14, 94–104, https://doi.org/10.1016/j.quageo.2012.03.015, URL `http://www.sciencedirect.com/science/article/pii/S1871101412000684`, 2012.

Trachsel, M. and Telford, R. J.: All age–depth models are wrong, but are getting better, The Holocene, 27, 860–869, https://doi.org/10.1177/0959683616675939, URL `https://doi.org/10.1177/0959683616675939`, 2016.

van Albada, S. and Robinson, P.: Transformation of arbitrary distributions to the normal distribution with application to {EEG} test–retest reliability, Journal of Neuroscience Methods, 161, 205 – 211, https://doi.org/10.1016/j.jneumeth.2006.11.004, 2007.

Van Huffel, S.: Total least squares and errors-in-variables modeling: bridging the gap between statistics, computational mathematics and engineering, pp. 539–555, Physica-Verlag HD, https://doi.org/10.1007/978-3-7908-2656-2_44, 2004.

Van Huffel, S. and Vandewalle, J.: The Total Least Squares Problem: Computational Aspects and Analysis, vol. 9 of *Frontiers in Applied Mathematics*, SIAM, Philadelphia, PA, 1991.

VanderPlas, J. T.: Understanding the Lomb–Scargle Periodogram, The Astrophysical Journal Supplement Series, 236, 16, https://doi.org/10.3847/1538-4365/aab766, URL `http://stacks.iop.org/0067-0049/236/i=1/a=16`, 2018.

Virtanen, P., Gommers, R., Oliphant, T. E., Haberland, M., Reddy, T., Cournapeau, D., Burovski, E., Peterson, P., Weckesser, W., Bright, J., van der Walt, S. J., Brett, M., Wilson, J., Millman, K. J., Mayorov, N., Nelson, A. R. J., Jones, E., Kern, R., Larson, E., Carey, C. J., Polat, İ., Feng, Y., Moore, E. W., VanderPlas, J., Laxalde, D., Perktold, J., Cimrman, R., Henriksen, I., Quintero, E. A., Harris, C. R., Archibald, A. M., Ribeiro, A. H., Pedregosa, F., van Mulbregt, P., and SciPy 1.0 Contributors: SciPy 1.0: Fundamental Algorithms for Scientific Computing in Python, Nature Methods, 17, 261–272, https://doi.org/10.1038/s41592-019-0686-2, 2020.

Zhu, F., Emile-Geay, J., McKay, N. P., Hakim, G. J., Khider, D., Ault, T. R., Steig, E. J., Dee, S., and Kirchner, J. W.: Climate models can correctly simulate the continuum of global-average temperature variability, Proceedings of the National Academy of Sciences, 116, 8728, https://doi.org/10.1073/pnas.1809959116, URL http://www.pnas.org/content/116/18/8728.abstract, 2019.

---

## Author Comment (AC3) · 16 Nov 2020

Thank you to the reviewer for these comments. Please find reviewer comments in *italics* and authors' responses in **bold**.

*This ms is clearly written and makes a convincing case for standardizing paleo data for subsequent compilation studies. It shows a range of very helpful developments in compiling and compiling multiple records, enabling much more informed analyses than currently available. It is high time for the paleo-community to stop neglecting chronological uncertainties. I hope that this initiative will be taken up and used to make more robust paleoclimatic inferences.*

[Figure]

*Pending just the addition of a few references, I suggest accepting this manuscript for publication.*

**We thank the reviewer for their constructive comments, and respond to specific suggestions below.**

*On p2, please also cite some earlier papers on comparisons between records that take into account chronological uncertainties: doi:10.1177/0959683607075857 doi:10.1016/j.quascirev.2008.07.009 doi:10.1016/j.quascirev.2010.11.012*

**We will better reference the discussion of earlier work on chronological uncertainty, thank you for the suggestions of additional papers.**

*For the comparison of Hulu and GISP2, it might also be useful to cite doi:10.1002/jqs.1330 which calculates 'event probabilities' within time-windows for individual records, and then calculates probabilities for synchronous reactions between two records as the product of these probabilities for each time window.*

**Agreed. We will add discussion of this paper in our expanded discussion of the Hulu/GISP2 comparison.**

*For section 5.4, a citation to a recent compilation paper in Science might also be helpful here: doi:10.1126/science.aay5538.*

**Thank you for the suggestion, we will add discussion of this paper.**

*Details p16 line 5, significant caption Fig. 8: the median*

**We will fix these typos.**

---

## Author Comment (AC4) · 16 Nov 2020

Thank you to the reviewer for these comments. Reviewer comments below in *italics*, authors' replies in **bold**.

*GeoChronR represents a nice new addition to the field of paleoclimatology and sediment-based paleoenvironmental reconstruction by presenting what is essentially a 'one-stop shop' for age uncertainty analysis in paleoenvironmental sequences, and includes code for analyzing data calibrated via a variety of different methods (e.g. layer counting, U-Th, radiocarbon), etc. This removes the need to use different software for each type of proxy archive, and as the authors note, should speed up the analysis of*

*large ensembles of paleoclimatic data.*

**We thank the reviewer for their constructive and encouraging review. We respond to specific points below.**

*I think the paper is mainly fine as is, just some minor changes to highlight the flexibility of the package and hone some of the science examples presented would be good.*

*That being said, given that this system uses the LiPD file format, I think it would be useful for the authors to include explicit mention of other utilities that exist (e.g. in Python) for converting other file formats (e.g. text files from NOAA Paleoclimatology). Our community still has a lot of different 'standards' for data archiving floating around, so it would be nice to emphasize that this software package can be used on other data provided the authors are willing to convert files to the correct format.*

**Thank you for this recommendation. We will include a discussion of the utilities available to convert data to LiPD.**

*I find the incorporation of correlation analysis especially compelling.*

*In the examples, for instance in 5.2, the purported lack of correlation between GISP2 and Hulu is quite controversial as I'm sure the authors know. I would like a bit more discussion of the reasons for this - I think the two records can be very strongly linked, but still not show strong Pearson correlations. GISP and sometimes Hulu seems to show 'on' or 'off' values - e.g. the record is jumpy, akin to a Dirac delta function especially for some of the rapid millennial-scale Heinrich events. Pearson product moment correlation, even if spectrally filtered, might not be appropriate, since it focuses on linear association. Would a ranked correlation metric like Spearman's rho show different results? Incorporating alternate metrics of correlation into GeoChronR is in my opinion not needed for this release, but it could be useful to mention here, along with the suggestion that users can use the age model output to build their own analyses using alternate correlation metrics, different assumptions about the underlying distributions*

*of the data. I imagine that is the hope anyways with such a flexible package.*

**Thank you for these comments and suggestions. We chose to illustrate the GISP2-Hulu correlation in the manuscript largely because it's an iconic age-uncertain relationship in paleoclimatology that we expected would be of general interest to readers. We also anticipated that the result would indeed be some-what controversial, although that is not our intent. There is a vast literature on Greenland ice core and Asian speleothem relationships, and our simple analy-sis was not intended to weigh in on that discussion, only highlight the potential value of considering age uncertainties when comparing two independently dated chronologies. We will expand on our discussion of the implication of our results to highlight the many of the components that our analysis doesn't examine.**

**By default, and in this example, geoChronR "gaussianizes" data before correlat-ing them, reducing some of the impact of the "on-off" nature of the Hulu (and many speleothem records. However, including rank-based correlations methods is an excellent, and fairly straightforward suggestion, which we will add to the package and describe in the manuscript.**

*I may be wrong, but haven't the Greenland ice core chronologies been revised to create chronologies that produce a stronger relationship between greenland oxygen isotopes and the Hulu record?*

**Certainly a lot of work has been done since 2001 on the chronologies for both the ice cores and the speleothems, which may improve the correlations. We will mention this in our expanded discussion of the results.**

*I really like the analysis with the Arctic temperature database and hope a more sub-stantial paper on the meaning of these PCs is forthcoming. I wonder if PC2 would be more easily interpreted if varimax rotation or some method were applied.*

**We agree that the results are interesting and we're keen to explore the implica-**

**tions of these results.**